# First passage time analysis of spatial mutation patterns reveals sub-clonal evolutionary dynamics in colorectal cancer

**Magnus J. Haughey**[1☯*], **Aleix Bassolas**[1☯], **Sandro Sousa**[1], **Ann-Marie Baker**[2,3], **Trevor A. Graham**[2,3], **Vincenzo Nicosia**[1*], **Weini Huang**[1*]

**1** School of Mathematical Sciences, Queen Mary University of London, London, United Kingdom, **2** Centre for Evolution and Cancer, Institute of Cancer Research, London, United Kingdom, **3** Centre for Genomics and Computational Biology, Barts Cancer Institute, Barts and the London School of Medicine and Dentistry, Queen Mary University of London, London, United Kingdom

☯ These authors contributed equally to this work.
\* m.j.haughey@qmul.ac.uk (MJH); v.nicosia@qmul.ac.uk (VN); weini.huang@qmul.ac.uk (WH)

## Abstract

The signature of early cancer dynamics on the spatial arrangement of tumour cells is poorly understood, and yet could encode information about how sub-clones grew within the expanding tumour. Novel methods of quantifying spatial tumour data at the cellular scale are required to link evolutionary dynamics to the resulting spatial architecture of the tumour. Here, we propose a framework using first passage times of random walks to quantify the complex spatial patterns of tumour cell population mixing. First, using a simple model of cell mixing we demonstrate how first passage time statistics can distinguish between different pattern structures. We then apply our method to simulated patterns of mutated and non-mutated tumour cell population mixing, generated using an agent-based model of expanding tumours, to explore how first passage times reflect mutant cell replicative advantage, time of emergence and strength of cell pushing. Finally, we explore applications to experimentally measured human colorectal cancer, and estimate parameters of early sub-clonal dynamics using our spatial computational model. We infer a wide range of sub-clonal dynamics, with mutant cell division rates varying between 1 and 4 times the rate of non-mutated cells across our sample set. Some mutated sub-clones emerged after as few as 100 non-mutant cell divisions, and others only after 50,000 divisions. The majority were consistent with boundary driven growth or short-range cell pushing. By analysing multiple sub-sampled regions in a small number of samples, we explore how the distribution of inferred dynamics could inform about the initial mutational event. Our results demonstrate the efficacy of first passage time analysis as a new methodology in spatial analysis of solid tumour tissue, and suggest that patterns of sub-clonal mixing can provide insights into early cancer dynamics.

**Data Availability Statement:** Data availability: Raw data analysed in this study are openly available and can be accessed at: https://doi.org/10.6084/m9.figshare.22182748.v1. Contact Queen Mary University of London Open Research team for any enquiries regarding data access (openresearch@qmul.ac.uk). Code availability: Scripts designed to generate and visualise sub-clonal mixing patterns are available at: https://github.com/MagnusHaughey/CancerSubclonalPatterns. Code relating to calculations of first passage time statistics can be found at: https://mygit.katolaz.net/covid_19_ethnicity/rw-segregation.

**Funding:** MJH is supported by the Life Sciences Initiative at Queen Mary University of London (https://www.qmul.ac.uk/). AB acknowledges funding from the Juan de la Cierva program (https://www.deusto.es/cs/Satellite/deusto/en/university-deusto/admissions-administration-and-grants/scholarships-and-grants-/juan-de-la-ciervaincorporacion-0/beca#:~:text=The%20Juan%20de%20la%20Cierva,first%20stage%20of%20postdoctoral%20training.); the Spanish Ministry of Universities; the European Union - Next Generation EU; the Recovery, Transformation and Resilience Plan (https://ec.europa.eu/info/strategy/recovery-plan-europe_en#nextgenerationeu); the University of the Balearic Islands (https://www.uib.eu/); the Departament d'Enginyeria Informatica i Matematiques, Universitat Rovira i Virgili, Tarragona, Spain (https://deim.urv.cat/), and Instituto de Fisica Interdisciplinar y Sistemas Complejos IFISC (CSIC-UIB), Campus UIB, 07122 Palma de Mallorca, Spain (https://ifisc.uib-csic.es/en/). VN acknowledges the support of the EPSRC New Investigator Award Grant No. EP/S027920/1 (https://www.ukri.org/councils/epsrc/). The funders had no role in study design, data collection and analysis, decision to publish, or preparation of the manuscript.

**Competing interests:** The authors have declared that no competing interests exist.

## Author summary

Tumours consist of a mosaic of cell sub-populations, which may differ in ways such as growth rate and their response to anti-cancer treatment. Understanding how sub-populations emerge and evolve is therefore central to predicting future behaviour of tumours and improving treatment regimes. Traditional methods of studying tumour heterogeneity, however, are often ignorant to the spatial context of cells, or preserve only low-resolution spatial data, yet spatial information may contain important clues about the evolution of tumour sub-populations. To exploit this fact, new methods of describing spatial patterns of tumour cells are needed in order to relate experimental observations to the underlying evolution of the tumour. In this study, we present a new methodology for spatial analysis of tumour tissue, utilising random walks to quantify clustering and heterogeneity within high-resolution spatial maps of tumour sub-population mixing. We apply our method to maps of sub-population mixing in colorectal cancer samples and, using spatial computational modelling, estimate the relative growth rate and age of mutated sub-populations, along with the strength of cell pushing within these samples. Our work demonstrates the potential of spatial information to inform about cancer evolution, and establishes a foundation for future research into spatial analysis of tumour data.

## Introduction

Understanding the origins and effects of intra-tumour heterogeneity is a fundamental challenge in cancer research and is central to characterising the evolutionary forces which drive disease progression and the response to treatment [1–5]. Tumours comprise of a diverse population of cells, often containing sub-populations, or *sub-clones*, carrying genetic alterations which may confer phenotypic changes such as increased rate of proliferation, elevated metastatic potential or evasion of the body's immune system. Understanding the evolution of sub-clones within tumours could therefore greatly advance our ability to perform better prognoses and design more effective anti-cancer treatment strategies [6].

Direct observation of the early sub-clonal evolution is often infeasible as, in some cancers, sub-clones arise in the early, undetectable, malignancy [4, 7]. This problem is confounded by the difficulty of obtaining temporal samples in humans. Recently, computational and mathematical modelling, however, has been applied to explore various questions including estimating the probability of developing treatment-resistant sub-clones, and characterising the influence of tissue architecture on the evolution of both neutral and oncogenic mutations [8–11]. Spatial information itself, often derived using histopathological approaches, is used in cancer diagnosis, classification and prognosis [12–14]. For example, the spatial distribution of tumour infiltrating immune cells has been shown to have prognostic value in some cancers [15–18] and, recently, higher dimensional measurements of tumour tissue obtained using CODEX technology have revealed correlations between the spatial arrangement of the tumour microenvironment and patient survival in colorectal cancer [19]. In this study, we reason that the sub-clonal spatial pattern is a readout of the evolutionary history of a tumour and could present a route to quantifying the evolution of sub-clones. To investigate this, we use spatial computational modelling to develop methodologies for inferring the growth history of tumour sub-clones based on their spatial arrangement in sampled tumours.

We apply our analysis to human colorectal cancer samples, where the spatial composition of point mutations is mapped at the cellular level using the BaseScope RNA *in situ*

hybridisation assay. Specifically, this technique is used to detect mRNA in mutant and non-mutant (wild-type, WT) cancer cells. Tumour tissue samples are combined with molecular probes, designed to bind specifically to the mutant and WT mRNA sequences of interest, resulting in signals which can be amplified, enabling a precise spatial mapping of tumour cells which are WT or mutated at that particular position in the genome. Previously, Baker *et al.* demonstrated the high sensitivity and specificity of this method, targeting a small panel of driver gene point mutations (within the *PIK3CA*, *BRAF* and *KRAS* genes) found commonly in human cancers [20].

In their study, Baker *et al.* used BaseScope to reveal diverse patterns of sub-clonal mixing between mutated and WT tumour cells (Fig 1a), and quantified these patterns using a spatial analogue of Shannon's entropy. Initial spatial analysis suggested that the observed BaseScope patterns were consistent with early arising, weakly selective, mutated sub-clones or later arising sub-clones endowed with a larger replicative advantage over the WT population. Whilst Shannon's entropy provided the first analysis of tumour heterogeneity, it is defined at a single spatial scale and thus is strongly scale dependent. We address this issue here by quantifying clustering and heterogeneity of cell mixing patterns at multiple spatial scales using the statistics of random walks to obtain a more complete description of the complex sub-clonal patterns elucidated with BaseScope.

Many spatial systems can be naturally represented as a network of interacting and connected nodes, of different classes, as a way to study the effects of the system's structure on its dynamics. Heterogeneity and segregation within networks can influence the statistics of a random walker on the graph, making random walks a useful tool in network science to quantify the structural properties of a system [21–24]. By quantifying the structure of the sub-clonal patterns at multiple length scales, therefore improving on previous analyses of these patterns, we propose that methods exploiting random walks have the potential to advance our understanding of the link between the spatial arrangement of sub-clones and the underlying sub-clonal dynamics.

In this study, we focus on the class mean first passage times (CMFPT) on a network, defined as the expected time, $\tau_{\alpha\beta}$, for a random walker beginning on a node of class $\alpha$ to first arrive at a node of class $\beta$. This method has recently been applied to other complex systems to quantify spatial segregation in voting patterns [25], ethnic segregation in UK and US metropolitan areas [26], and to show that internal clustering and spatial heterogeneity of individuals of different ethnic groups contributed to the observed excess of infectious diseases [27]. Here, we leverage the CMFPT to quantify the complex spatial patterns of mutated sub-clonal cells in BaseScope images.

We first demonstrate the capability of the CMFPT method to measure pattern structures using an artificial model of WT and mutant population mixing. We then compare CMFPT measurements of sub-clonal mixing patterns in human colorectal cancers to the same measurements derived from our agent-based simulations. We perform parameter estimation using our computational model, predicting the relative age and replicative fitness advantage of the mutated sub-clone and the strength of cell pushing most consistent with the spatial patterns observed with BaseScope. Our analysis indicates that sub-clonal populations often appear relatively early in the expansion of the WT population, and exhibit a wide range of fitness advantages over WT cells. This work demonstrates the capability of the class mean first passage time as a method of quantifying cell mixing patterns *in vivo*, and our findings suggest that patterns of sub-clonal mixing in mature tumours could potentially provide insights into early sub-clonal dynamics.

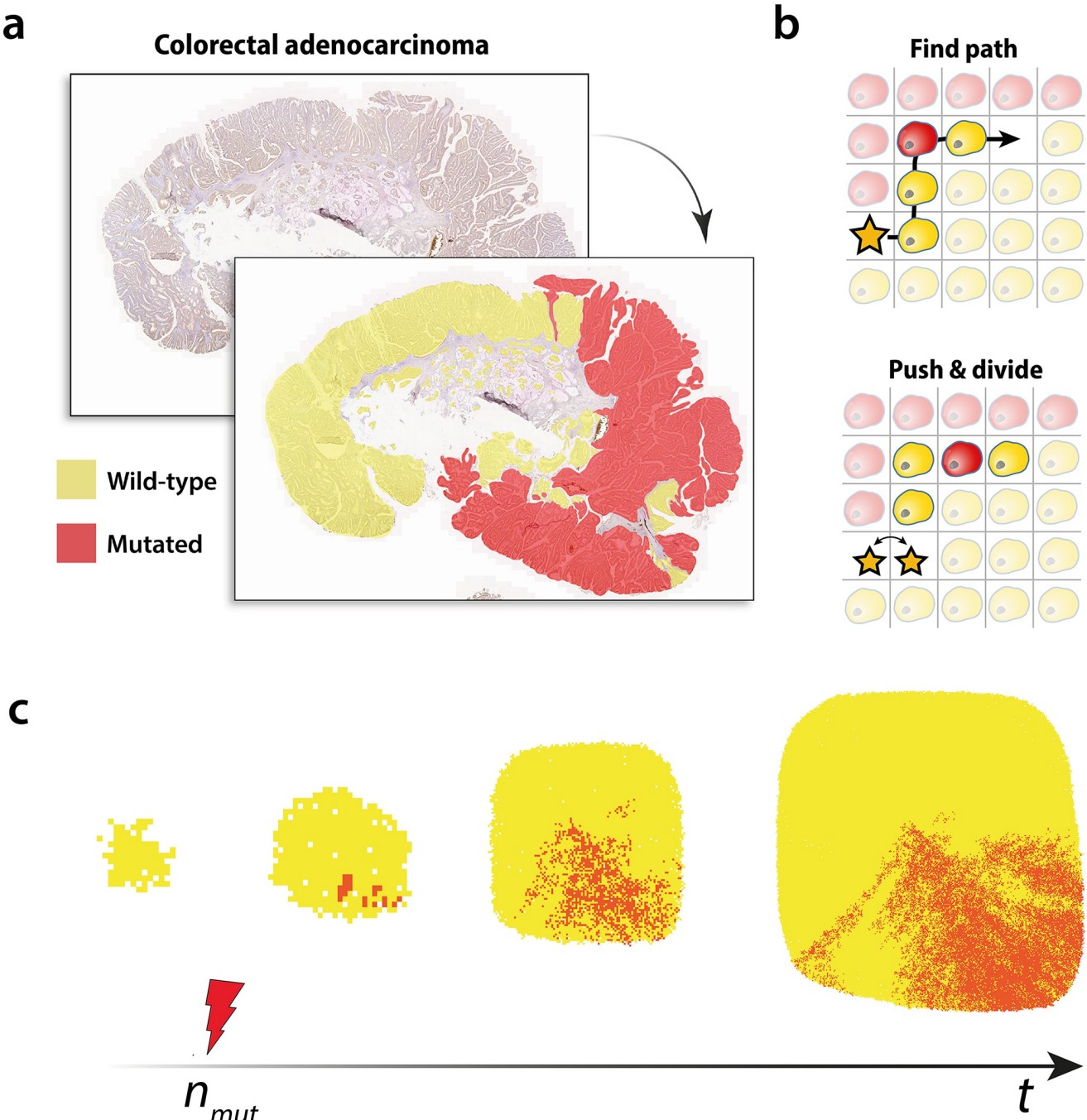

**Fig 1. Sub-clonal mixing in human colorectal tumours. (a)** Sub-clonal mixing patterns in human colorectal adenocarcinoma tissue revealed with BaseScope. Tumour cells carrying the *KRAS* G12A mutation are highlighted in red. Those which are wild-type at this loci are highlighted in yellow. **(b)** Cell pushing in spatial simulations. A dividing cell (star) creates the space needed to divide into two cells by pushing neighbouring cells along a path to a nearby empty lattice point. **(c)** Computational model of tumour sub-clonal mixing. Competing wild-type (yellow) and mutated (red) cell populations clonally expand on a 2-dimensional square lattice.

## Results

### Computing first passage time statistics

To estimate the class mean first passage time (CMFPT), we simulate the trajectory of random walkers on cell mixing patterns embedded in a square lattice. The mean number of steps, $\tau_{\alpha\beta}$, taken to first arrive at a node of class $\beta$ when departing from a node of class $\alpha$, is computed over all starting nodes of the same class $\alpha$ on the graph. In this study we focus on spatial patterns of two classes, mutated (red) and WT cells (yellow), giving rise to four possible CMFPT quantities: $\tau_{rr}$, $\tau_{ry}$, $\tau_{yr}$ and $\tau_{yy}$. Different spatial patterns will differ naturally in magnitude and shape. Thus, to compare the CMFPT of different images we normalise the first passage times for each image to the corresponding quantities derived under a null model, in which the constituent classes in the spatial pattern are reassigned uniformly at random to each lattice point, but maintaining the abundances of each class and the general geometry of the original image. We denote these quantities as the normalised first passage times $\tilde{\tau}_{rr}$, $\tilde{\tau}_{ry}$, $\tilde{\tau}_{yr}$ and $\tilde{\tau}_{yy}$.

We explore the phase space spanned by $\tilde{\tau}_{ry}$ and $\tilde{\tau}_{ry}/\tilde{\tau}_{yr}$, where $\tilde{\tau}_{ry}$ and $\tilde{\tau}_{yr}$ denote the normalised CMFPT for red→yellow and yellow→red transitions respectively. Through our analysis we find that the combination of these quantities enables us to distinguish between spatial patterns which differ in heterogeneity and pattern structure, and separates images containing clusters of different characteristic sizes. Measurements in this phase space have been used in previous analyses involving CMFPT to characterise colour distributions on 2-dimensional lattices [25]. Spatial patterns in which the two colours are distributed uniformly at random will lie at $(\tilde{\tau}_{ry}, \tilde{\tau}_{ry}/\tilde{\tau}_{yr}) = (1, 1)$, since this particular pattern exactly resembles the null model used for normalisation. In general, patterns which contain a more ordered, or segregated, distribution of colours give rise to $\tilde{\tau}_{ry} \gg 1$. The CMFPT reflects not only the extent of segregation of the colours, but also the fine-grain details of the patterns. As a result, the size, shape and spatial distribution of clusters will affect the CMFPT, with patterns containing large yellow clusters leading to $\tilde{\tau}_{ry}/\tilde{\tau}_{yr} < 1$ and those with large red clusters giving rise to $\tilde{\tau}_{ry}/\tilde{\tau}_{yr} > 1$.

We define the class ratio, $\phi$, of an image as the ratio of red to yellow classes,

$$\phi = \frac{N_r}{N_r + N_y},$$

(1)

where $N_r$ and $N_y$ represent the number of red and yellow pixels in an image respectively. Note while $\phi$ is a direct input in our artificial cell mixing model, it is an outcome of the agent-based simulations, depending on the replicative advantage of the sub-clonal population, the relative time at which it emerges in the simulations, and the strength of cell pushing.

### A model of 2-dimensional cell mixing patterns

We first develop a simple model to generate a set of diverse spatial patterns of two colours (red and yellow) in a 2-dimensional lattice background. In particular, we generate three distinct patterns, which are: (1) clusters; (2) large centred cluster and (3) column arrangement (Fig 2a–2f). We run a number of simulations whilst varying class ratio, $\phi$, between $0 < \phi < 1$ for each of these patterns and calculate the corresponding normalised CMFPT in the $(\tilde{\tau}_{ry}, \tilde{\tau}_{ry}/\tilde{\tau}_{yr})$ phase space in each case (Fig 2g). Results from these initial calculations demonstrate how pattern structure and class ratio are reflected in the CMFPT, with a clear correspondence between $\phi$ and $\tilde{\tau}_{ry}/\tilde{\tau}_{yr}$ value (vertical axis) observed across all patterns. Patterns with $\phi < \frac{1}{2}$ generally lie below $\tilde{\tau}_{ry}/\tilde{\tau}_{yr} = 1$, and those with $\phi > \frac{1}{2}$ fall above $\tilde{\tau}_{ry}/\tilde{\tau}_{yr} = 1$, reflecting the increasing segregation of classes at either extreme of class ratio. Moreover, this effect is intensified for specific

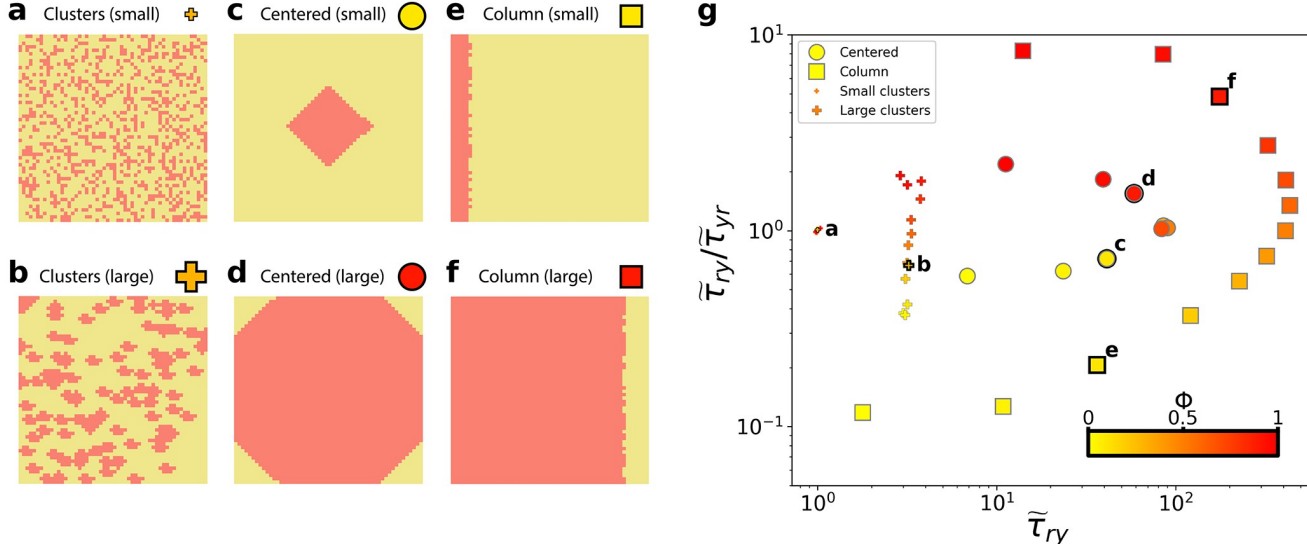

**Fig 2. Characterising 2-dimensional mixing patterns using CMFPT.** Patterns were generated using the **(a, b)** clusters; **(c, d)** centred and **(e, f)** column models. **(g)** Location of the patterns in the $(\tilde{\tau}_{ry}, \tilde{\tau}_{ry}/\tilde{\tau}_{yr})$ phase space obtained for three models with varying class ratio, $\phi$.

spatial patterns, with a stronger dependence for fully segregated patterns (*i.e.* column patterns *e, f* in Fig 2g), and a weaker dependence for more intermixed patterns (*i.e.* cluster patterns *a, b* in Fig 2g).

Our normalised CMFPT measure allows for more precise interrogation of pattern structures than class ratio alone. The separation of each pattern type is shown along the $\tilde{\tau}_{ry}$ (horizontal) direction of Fig 2g. Being most similar to its corresponding null model used to normalise the CMFPT, the small clusters pattern (Fig 2a) occupies a region of the phase space near to $(\tilde{\tau}_{ry}, \tilde{\tau}_{ry}/\tilde{\tau}_{yr}) = (1, 1)$. The normalised CMFPT measurement is sensitive to an increase in the cluster size, even when class ratio remains constant (Fig 2b). For both the centred (Fig 2c and 2d) and column (Fig 2e and 2f) patterns, discrete trajectories in the phase space are observed in Fig 2g, with the value of $\phi$ determining the position of the specific pattern along the curve.

We compared the CMFPT metric to Shannon's entropy (S1 and S2 Figs), which was previously applied by Baker *et al.* to colorectal cancer samples analysed with BaseScope [20], and mean shortest distance, a similar measure to the CMFPT (see Methods). Shannon's entropy, $H$, ranges between $H = 0$ for fully segregated patterns, and $H = 1$ for fully mixed patterns (S1 (a) & S1(b) Fig) and, when supplemented with information on $\phi$ for each pattern, is able to describe some of the geometrical features of the different pattern structures. Crucially, however, we found that it is not sensitive to changes in the column model, and is only weakly sensitive to changes in patterns generated for different centered cluster sizes (S1(i) Fig). More fundamentally, the Shannon's entropy approach requires one to choose a scale at which to analyse the pattern. This should not affect the analysis of the simple patterns analysed here, since these patterns typically contain structures with only one, known, characteristic length scale, however it limits the efficacy of Shannon's entropy when applied to more complex data, such as biological tissue. Indeed, more complex data such as that derived from human tissue may exhibit multiple characteristic length scales which are not known *a priori*, and the size and dimensions of samples cannot be reliably controlled at all stages of the data acquisition process.

Analysis of the same patterns using the mean shortest distance metric produces similar results to those derived using the CMFPT (S3 and S4 Figs). As with the CMFPT, the mean shortest distance is computed for transitions between cell types, such as the yellow to red mean shortest distance, $\tilde{d}_{yr}$, and is related the mean shortest path, in terms of number of nodes, connecting cells of one type to the other. The quantity itself is normalised, similar to the CMFPT, by the corresponding quantity derived in a null model in which the constituent classes in the spatial pattern are reassigned uniformly at random to each lattice point (see Methods). Although CMFPT and mean shortest distance are similar measures, the main difference between them is in the construction of the path between nodes. Mean shortest distance is a deterministic measure of the distance between cells, whereas CMFPT constructs paths between pairs of cells using a stochastic process, resulting in a distribution of first passage times for each starting cell. As such, CMFPT may provide different information about the texture of the patterns than mean shortest distance by incorporating information about the neighbourhood of each cell. In this study, we summarise the CMFPT distributions using the mean, however one could also make use of higher order moments of the distributions to obtain further information about each cellular neighbourhood.

## Analysing simulated tumour sub-clone mixing patterns

We next investigate the efficacy of our method in characterising cell mixing patterns generated *in silico*. We developed a generic 2-dimensional agent-based simulation model of expanding tumour populations, incorporating cell birth, death and mutation on a lattice (Fig 1b and 1c). Simulations begin with a single WT tumour cell in the centre of the lattice which seeds a growing WT population. After a threshold number of WT cell divisions, specified by the model parameter $n_{mut}$, an existing WT cell acquires a mutation which confers a fitness advantage of $s$ (see Methods). We model fitness via a modulation of cell replication rates, such that cells carrying the mutation have a replication rate which is $(s \times 100)\%$ greater than that of WT cells. Neutral selection, where the mutant sub-clone grows at the same rate as the WT population, corresponds to $s = 0$. Here, $n_{mut}$ represents the fraction of the final system size, $N_{max}$ (a fixed model parameter), at which point the mutant population first appears. Throughout this paper we will refer to this parameter as representing the time of sub-clone emergence, however this interpretation should be adopted with caution. Despite an increase in $n_{mut}$ implying a later arising mutation, the number of cell replication events per unit time scales with the size of the tumour and thus $n_{mut}$ should not be interpreted straight-forwardly as a time, but rather an expression of absolute number of cell divisions. For example, for $N_{max} = 10^5$ and $n_{mut} = 0.1$, the mutation appears in an existing WT cell once the initially expanding WT population has reached $N_{max} \times n_{mut} = 10^4$ cells. All descendants of the mutated cell also carry the mutation. These simulations result in 2-dimensional spatial patterns of WT cells coexisting with a subclonal population of mutated cells. We utilise this framework to explore the impact on the resulting patterns of population mixing when varying three model parameters, i.e. mutant replicative advantage, $s$, the relative time, $n_{mut}$, at which the mutation appears in the growing WT population, and the strength of cell-cell pushing on the lattice, $q$ (see S1 Table).

Before analysing the patterns of the normalised CMFPT as a function of $s$, $n_{mut}$ and $q$, we first characterise its relationship with the class ratio, $\phi$, in each of the patterns (Fig 3). Similar to our observation in the artificial cell mixing model, we find a complex non-linear relationship between $\phi$ and the CMFPT, and a general trend of increasing $\tilde{\tau}_{ry}/\tilde{\tau}_{yr}$ with class ratio. In the artificial model, we can vary $\phi$ and the underlying pattern independently, since $\phi$ is an explicit model parameter. In our agent-based tumour model, however, this ratio has a dependence on $s$, $n_{mut}$ and $q$, being simply a readout quantity rather than a tunable parameter. Thus,

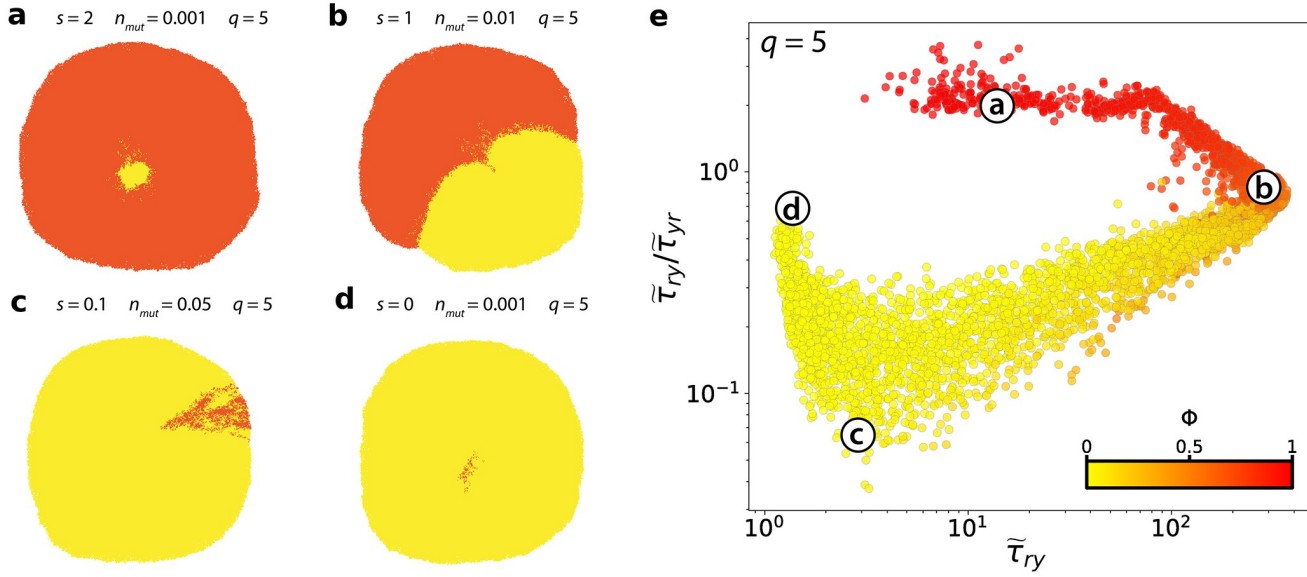

**Fig 3. Analysis of simulated sub-clonal mixing patterns. (a-d)** Simulated sub-clonal mixing patterns with cell pushing strength of $q = 5$. **(e)** Analysis of all simulated mixing patterns simulated with a cell pushing value of $q = 5$ in the $(\tilde{\tau}_{ry}, \tilde{\tau}_{ry}/\tilde{\tau}_{yr})$ phase space. Data represent simulations for all possible combinations of $s$ and $n_{mut}$ where $s \in \{0, 0.1, 0.2, 0.5, 1, 2, 3\}$ and $n_{mut} \in \{0.001, 0.01, 0.03, 0.05, 0.08, 0.1, 0.5\}$, with $q = 5$ (approximately 100 simulated patterns for each parameter combination). Point colour corresponds to the relative abundance of red and yellow, $\phi$.

class ratio and pattern structure are naturally coupled. Correspondingly, certain underlying pattern structures are observed only for small mutant sub-clones, and therefore small $\phi$ (Fig 3d), whereas combinations of $(s, n_{mut}, q)$ which give rise to higher $\phi$ will also result in substantially different pattern topologies (Fig 3a). As a result, the CMFPT measurements of the simulated patterns do not fall onto distinct curves (Fig 3e), as was observed for different underlying structures in the artificial model.

Beyond class ratio, we investigate the extent to which the CMFPT reflects spatial patterns generated under different model parameter combinations (Fig 4). When we classify our simulations according to mutant fitness, $s$, and the arising time of the mutant, $n_{mut}$, spatial patterns generated by different parameter combinations occupy distinct regions of the $(\tilde{\tau}_{ry}, \tilde{\tau}_{ry}/\tilde{\tau}_{yr})$ phase space, demonstrating that the CMFPT can be used to quantify distinct spatial heterogeneity. Patterns corresponding to neutral selection, $s = 0$ (purple points in Fig 4), lie in the leftmost region $1 \leq \tilde{\tau}_{ry} \leq 2$, and sub-clones with strong selection, $2 \leq s \leq 3$ (yellow and red points in Fig 4), are associated with larger values of $\tilde{\tau}_{ry}$. Due to the coupled effects of $s$ and $n_{mut}$, tumours with low $s$ and $n_{mut}$ (Fig 4, navy & turquoise circles), and those with large $s$ and $n_{mut}$ (Fig 4, yellow & orange triangles) fall into similar regions of the phase space, especially for $q = 0$. Here, the CMFPT may struggle to distinguish these different dynamics. Nevertheless we would still expect to see differences in the appearance of the sub-clonal patterns under these different dynamical regimes, however, and it does appear that the mean CMFPT of these patterns subtly reflects these differences. Tumours with the earliest arising sub-clones with the strongest replicative advantage (Fig 4, red & orange circles), representing the most extreme dynamics we tested, occupy a unique area of the phase space for all cell pushing strengths.

Varying the strength of cell pushing, $q$, has clear qualitative impacts on spatial heterogeneity. At weak pushing strengths cell displacement occurs at short ranges, leading to greater clustering of sub-clones. Conversely, at larger values of $q$ cells are subject to more frequent displacement by nearby dividing cells, so that any sub-clonal clusters are more quickly

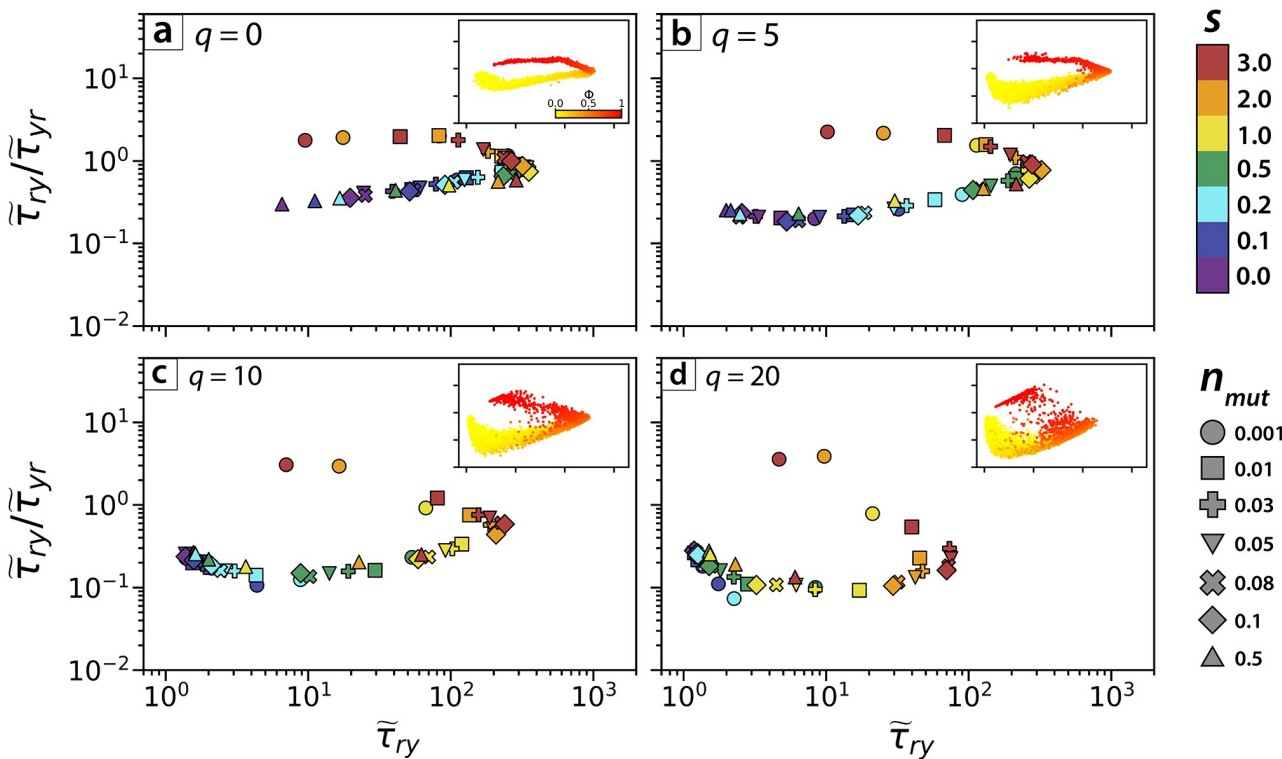

**Fig 4. Analysis of simulated sub-clonal mixing patterns for varying sub-clonal dynamics.** Mean CMFPT values for simulated sub-clonal mixing patterns in the phase space $(\tilde{\tau}_{ry}, \tilde{\tau}_{ry}/\tilde{\tau}_{yr})$, with points coloured according to values of model parameter $s$ and point shape depending on parameter $n_{mut}$. Data within each panel represent simulations for all possible combinations of $s$ and $n_{mut}$ where $s \in \{0, 0.1, 0.2, 0.5, 1, 2, 3\}$ and $n_{mut} \in \{0.001, 0.01, 0.03, 0.05, 0.08, 0.1, 0.5\}$ (approximately 100 simulated patterns for each parameter combination). Inset plots contain the same data as in the main panels, but show individual data points (one per pattern, $n \approx 100$ patterns per $s$ and $n_{mut}$ combination) with point colour corresponding to the class ratio, $\phi$. Images are separated depending on their pushing value $q = 0$ **(a)**, $q = 5$ **(b)**, $q = 10$ **(c)** and $q = 20$ **(d)**.

dispersed. These effects are also reflected in the distribution of CMFPT measurements in the $(\tilde{\tau}_{ry}, \tilde{\tau}_{ry}/\tilde{\tau}_{yr})$ phase space (Fig 4a–4d). For weak pushing strengths (Fig 4a) the simulated spatial patterns follow a path in phase space more similar to that of the centred artificial model patterns (Fig 2c and 2d), with two distinct "arms" demarcating patterns with $\phi < 0.5$ and those with $\phi > 0.5$. In the limit of zero cell pushing ($q = 0$), varying the mutant replicative advantage, $s$, and arising time, $n_{mut}$, impacts the final class ratio. Since, however, cell proliferation is limited to the periphery of the system, we observe little variation in the size and spatial heterogeneity of sub-clonal patterns for different realisations of the same system (*i.e.* the same value of $s$ and $n_{mut}$). This lack of variation in sub-clonal pattern structure at weaker $q$ is reflected in the CMFPT, with measured values clustering strongly in the phase space according to the values of $s$ and $n_{mut}$ and the class ratio of the sub-clonal pattern (Fig 4a, inset).

Conversely, under stronger cell pushing the diversity in the observed sub-clonal pattern structures is greater, resulting in increased variation in measured CMFPT values for different tumours with the same $s$ and $n_{mut}$ values, and weaker clustering of CMFPT measurements according to the class ratio of the simulated pattern. This is explained by the loss of distinct clusters of closely related cells when cell pushing strength is increased. The increased fragmentation of sub-clones is captured by the CMFPT and represented by a steeper initial decline in $\tilde{\tau}_{ry}/\tilde{\tau}_{yr}$ for patterns with low $s$.

Overall, our analysis of simulated tumour sub-clonal mixing patterns points to an important result. That is, despite carrying little genetic information, different underlying sub-clonal growth dynamics each have their own spatial signature, left behind in the mixing patterns of WT and sub-clonal tumour cell populations, and it is possible to narrow down the potential underlying cell dynamics by quantifying the resulting spatial patterns alone. Moreover, our results suggest that CMFPT provides a promising new route to understanding these spatial patterns by describing the structural features of the patterns over multiple spatial scales.

## Quantifying sub-clonal dynamics in human colorectal cancer

Having shown that CMFPT can be used to recover the underlying parameters of generative models by analysing spatial mutation patterns alone, both with a simple artificial model and agent-based simulated tumours, we next explore an application of our method to human colorectal carcinoma samples spatially mapped by BaseScope [20]. We analysed 22 cell mixing patterns produced with BaseScope (S10(a) Fig) which detail the spatial composition of tumour cells, mapping out sub-clonal populations carrying point mutations in one of three genes commonly mutated in colorectal cancer—*PIK3CA*, *BRAF* and *KRAS*. Mutations in these genes have been associated with cancer driver events, poor prognosis and treatment resistance [28–32]. Based on our understanding of how the CMFPT measurement relates to sub-clonal mixing patterns generated with our agent-based model, we aim to reconstruct the underlying dynamics driving the growth of the mutated sub-clones observed with BaseScope, in terms of our computational model parameters. It is important to note that, whilst the mutation time in our 2-dimensional simulations, $n_{mut}$, describes the time at which the mutation first appears in the entire system, the interpretation of this parameter changes when applied to the BaseScope samples. It is extremely rare that the same mutation will happen independently in multiple regions of any tumour, yet since the BaseScope images represent 2-dimensional sub-samples of a larger 3-dimensional system, a single spatially continuous sub-clone in the 3-dimensional tumour can lead to the appearance of spatially distinct sub-clonal regions in 2-dimensional samples. Accordingly, any inferred values of $n_{mut}$ will represent the relative length of time that the WT population was expanding in that localised area prior to the mutant sub-clone. In a few cases, where the BaseScope samples involve multiple spatially discontinuous tumour areas, the distribution of inferred $n_{mut}$ values across all local areas reveals how fast the mutant population spread in these tumours (details explained below).

Whilst the number of tumour cells contained within the experimental colorectal cancer samples is of the same order of magnitude as the simulated patterns to which they will be compared, the experimental images vary in resolution and contain far more pixels than the simulated patterns. Furthermore, we fix the final number of cells in our simulations, however the number of cells contained within the experimental patterns, and smaller sub-regions which we later analyse, varies across samples. Despite our efforts to parameterise our model such that simulated and experimental patterns contain a similar number of tumour cells, our comparisons may not be optimal in this regard and could possibly be more informative by improving the concordance of tumour cell numbers under comparison. A modified agent-based model, in which the final system size, $N_{max}$, is allowed to vary, could allow for better matching of cell numbers. Alternatively, keeping $N_{max}$ fixed in the simulations, but extracting sub-regions of the simulated patterns, the size of which could be varied, could also achieve this. One of the strengths of the CMFPT, however, is the normalisation of this quantity to corresponding first passage times in a null model, which enables the comparison of geometrical features of patterns of different sizes.

Initial analyses of the BaseScope images, for which we estimated the CMFPT over the totality of each image, highlighted some important additional processing steps that should be applied to the BaseScope images prior to parameter inference. Crucially, the presence of un-highlighted areas in the BaseScope images, representing all other non-cancerous tissue, significantly impacted our estimations of the CMFPT. We do not consider non-cancerous cells in our agent-based model, in order to maintain modest model complexity, however this makes naive comparisons of experimental to simulated sub-clonal patterns less informative. This is evidenced by our initial analysis of the unprocessed BaseScope images where we found some CMFPT values falling far outside of the range of measured times from our simulated patterns, and little correlation between the CMFPT of a BaseScope pattern and it's class ratio (S11 Fig). To improve comparisons between experimental and simulated sub-clonal patterns, we execute a number of pre-processing steps on the BaseScope images. We first process each image by manually filling in un-highlighted interior areas of colonic crypts in predominantly WT or mutant regions with the relevant colour (S10(b) Fig). To address the remaining un-highlighted areas which correspond to regions of non-cancerous cells (*e.g.* connective tissue), we separate BaseScope images containing spatially discontinuous tumour regions into smaller sub-sections. These sub-sections are not arbitrarily selected, but instead are manually delineated regions, identified by us, which contain locally connected areas of WT and mutated cells. By analysing these sections individually, we reduce the influence of un-highlighted space separating these regions, enabling a better comparison of experimental and simulated measurements.

Following these pre-processing steps, the results are largely improved compared to our initial analyses of the full BaseScope images, and reveal a similar general trend to that observed with our *in silico* pattern results, where higher $\phi$ patterns appear in the $\tilde{\tau}_{ry}/\tilde{\tau}_{yr} \geq 1$ region of the phase space, and lower $\phi$ patterns are situated below $\tilde{\tau}_{ry}/\tilde{\tau}_{yr} = 1$ (Fig 5e). Only a handful of the analysed BaseScope patterns lie below $\tilde{\tau}_{ry}/\tilde{\tau}_{yr} = 0.5$, a region of the phase space predominantly occupied by simulated patterns with high cell pushing, $q$, and low mutant selection, $s$, suggesting cell pushing might be generally weak *in vivo*.

To describe the sub-clonal dynamics more quantitatively, we performed a grid-search to find the best-fit simulated parameters for each of the analysed sub-components in the Base-Scope images (Fig 5a–5d, S12–S27 Figs) (see Methods). Plotting the best-fit simulated patterns demonstrates that our method not only captures the statistical features of the experimental images, but that the best-fit simulated patterns often also share a close visual correspondence to the BaseScope patterns both in terms of class ratio and pattern heterogeneity. BaseScope patterns with high segregation of WT and mutant populations are predicted to have low cell pushing. Highly segregated patterns which, in addition, have a large mutant frequency, $\phi$, are most consistent with model patterns simulated with early arising mutant sub-clones endowed with a strong replicative advantage.

We tested the robustness of our inference method by predicting model parameter values for simulated sub-clonal patterns, generated with known values of $s$, $n_{mut}$ and $q$ (Fig 5f). In the phase space of first passage times we find that the spread of values for simulated tumours with identical parameters can be reasonably large, and that the distributions obtained for tumours differing only by small changes in their parameters are not always easily distinguishable. Despite this, we find our method of inference to be reasonably accurate for parameters $s$ and $q$, as evidenced by the agreement between the "ground truth" parameter values and the predicted values. Large values of $s$ were more consistently correctly predicted than lower values, which is likely due to the small differences between values at the low end of the parameter range, resulting in sub-clonal patterns with similar statistical properties which were then misclassified by our algorithm. Of the three model parameters of interest, our method appears to be least

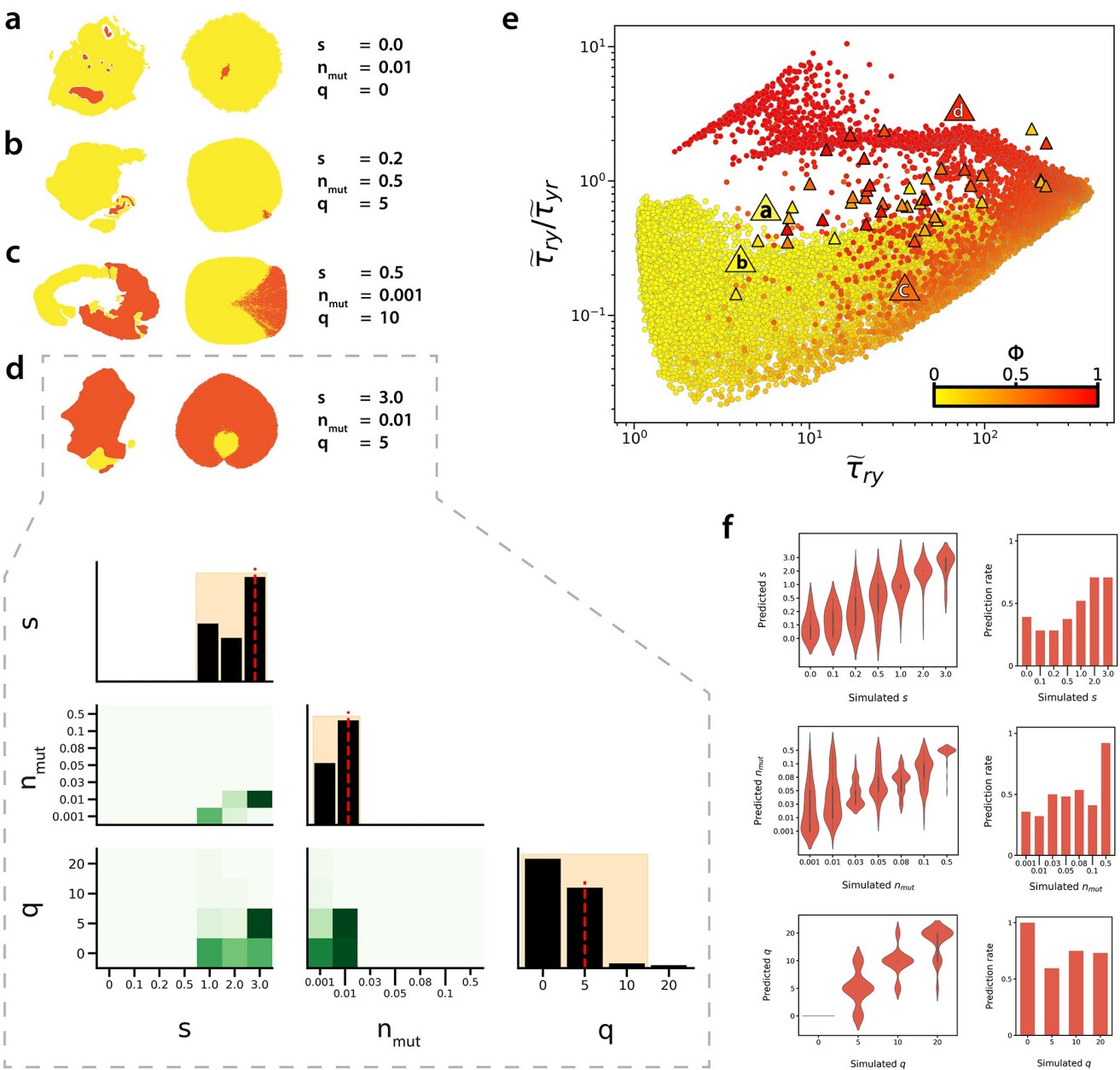

**Fig 5. Analysing sub-components of BaseScope sub-clonal patterns. (a-d)** Representative examples of sub-components of the BaseScope samples (after pre-processing steps applied) and their best-fit simulated sub-clonal patterns. **(d)** Marginal posterior distributions of model parameters $s$, $n_{mut}$, and $q$, comprising of the $n = 100$ nearest simulated sub-clonal patterns in the 4-dimensional phase space spanned by $\tilde{\tau}_{rr}$, $\tilde{\tau}_{ry}$, $\tilde{\tau}_{yr}$ and $\tilde{\tau}_{yy}$. Inferred parameter value is indicated by the vertical dashed line in the diagonal panels. 95% credible regions lie within the shaded region in the diagonal panels. **(e)** CMFPT analysis of all simulated sub-clonal patterns (circles) and connected WT and mutant cell sub-regions within BaseScope images (triangles). Points are coloured according to pattern class ratio, $\phi$. **(f)** Distribution of predicted $s$, $n_{mut}$, and $q$ values inferred from simulated patterns (left) and probability of correct prediction for each parameter value (right).

sensitive to the timing of the mutation, $n_{mut}$. Patterns simulated with intermediate values ($n_{mut} = 0.01, 0.03, 0.05, 0.08$ & $0.1$) are often predicted to have a $n_{mut}$ value an increment above or below the true value (*e.g.* true value of $n_{mut} = 0.05$ ascribed a value of $n_{mut} = 0.03$ or $n_{mut} = 0.08$). These values, however, correspond to differences of 2,000–3,000 WT cell divisions, which are small relative to the size of the whole tumour itself. Accuracy with respect to cell

pushing strength, $q$, was particularly good, with our method appearing to correctly predict the value of $q$ in the majority of cases.

The inferred best-fit parameters for connected WT and mutant regions within our Base-Scope images point to a wide range of sub-clonal growth dynamics. Analysis of the majority of the BaseScope samples suggests that the mutated populations experience some degree of replicative advantage, $s$, over the WT population, with some patterns consistent with mutated cells dividing at up to 4 times the rate of WT cells ($s = 3$). Approximately 10% of the sub-regions we analysed were consistent with neutral sub-clonal dynamics ($s = 0$). Inferred values for the relative time of mutant sub-clone invasion, $n_{mut}$, within each BaseScope sub-component range between 0.1% and 10% for the majority of the samples, suggesting that in many of the sub-regions we analysed, the mutant sub-clones arose early in the expansion of the WT tumour populations. Consistent with our earlier supposition, the inferred value of cell pushing parameter $q$ is mostly $q = 0$ or $q = 5$, however some larger inferred pushing values are predicted.

The newly implemented pre-processing steps reduced the influence of spatial regions of non-tumour tissue on the CMFPT measurements, improving the concordance between experimental and simulated tumours. Yet, not all of the experimental samples are visually consistent with their ascribed best-fit model image, despite having similar CMFPT values. This is due to the variance in the distribution of CMFPT values for certain combinations of $s$, $n_{mut}$ and $q$, meaning that, in some instances, experimental patterns can have CMFPT values consistent with multiple combinations of model parameters (S9 Fig). As a result, we occasionally assign best-fit model patterns and parameters that do not accurately resemble the experimental image (for example, sub-region 1 in Fig 6a), despite being supported statistically within the inference framework.

For three of the most fragmented BaseScope images, samples 02, 28 and 34, we were able to separately analyse several isolated sub-components of tumour cells and construct a distribution of inferred parameter values across the full images (S28 and S29 Figs & Fig 6 respectively). We surmise that the distribution of inferred selection strength, $s$, and mutation time, $n_{mut}$, can offer additional insights into the nature of the early sub-clonal evolution in these tumours. As previously mentioned, $n_{mut}$ should not be interpreted as the time at which the mutant population first appears in the tumour, but instead it indicates the relative length of time that the WT population was expanding in that area prior to invasion by the expanding mutated sub-clone. Despite this, there will be some correspondence between the distribution of inferred local $n_{mut}$ values and the time of the original mutational event: with our computational modelling demonstrating that earlier arising mutations tend to lead to higher infiltration of the mutant sub-clone throughout the mature tumour, and less variegation in the mutant sub-clone pattern. Following this rationale, we would expect an earlier mutational event to result in a unimodal distribution of inferred $n_{mut}$ values, with a low variance. Conversely, later occurring mutations should lead to a wider distribution of inferred $n_{mut}$ values, which may deviate from unimodality.

In particular, sample 34 (Fig 6) suggests high mutant replicative advantage and an early emerging sub-clone. This sample possesses a narrow distribution of inferred $n_{mut}$ across the different sub-sampled regions, concentrated towards low $n_{mut}$ values, indicating early invasion of the mutant sub-clone within the WT population, and pointing to an overall early emerging mutation in the evolution of the tumour. This narrow distribution of $n_{mut}$ is coupled to a narrow distribution of $s$, trending towards large values. Given that our inference method is most accurate when predicting large selection strengths (Fig 5f), this trend of high selection across the majority of the sub-regions would suggest that the mutant sub-clone as a whole is indeed subject to strong selection.

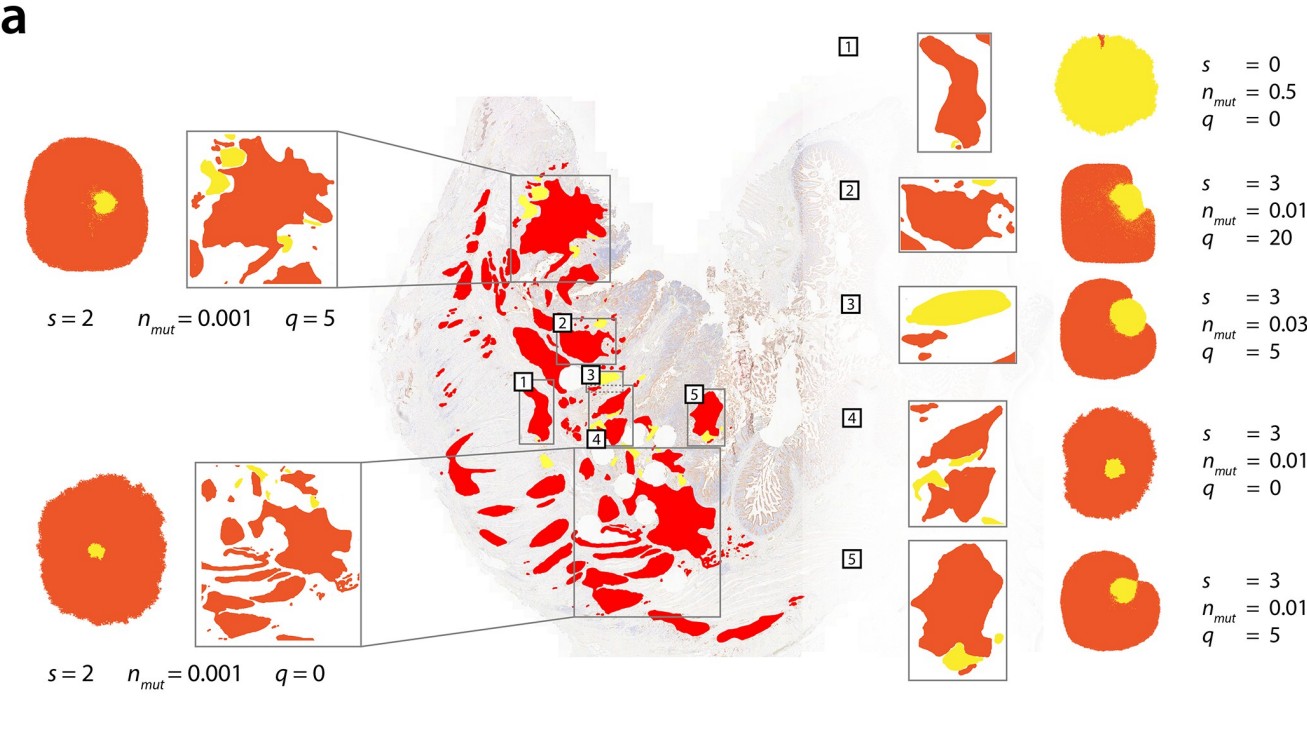

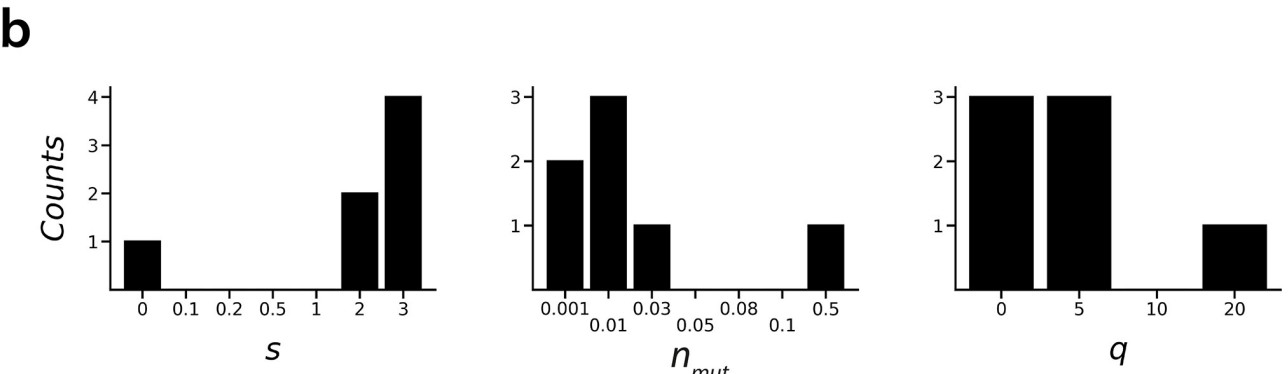

**Fig 6. Sub-section analysis of BaseScope sample 34. (a)** Best-fit simulated sub-clonal pattern shown next to each sub-section with corresponding model parameters. **(b)** Marginal distributions of inferred model parameters across all sub-sections.

These factors point to an early arising, positively selected mutant sub-clone within this tumour. Conversely, samples 02 and 28 (S28 and S29 Figs) contain a larger proportion of sub-regions consistent with a late invading mutant sub-clone ($n_{mut} = 0.5$), and a wider distribution of inferred $n_{mut}$ overall, along with lower inferred selection strength. Despite our inference method appearing to have reduced accuracy for intermediate $n_{mut}$ values, the results from our inference validation lend support to the wide distribution of sub-clone invasion times which were predicted within samples 02 and 28. Whilst we have limited resolution in these distributions, this might suggest that the mutation itself occurred later in the evolution of samples 02 and 28 than it did in sample 34.

Our inference of cell pushing within samples 02, 28 and 34 pointed to a trend of weak cell pushing. One should take care, however, when drawing conclusions about the cell pushing

strength within these samples as a whole based on the sub-region analyses. This is because, of the three model parameters we inferred, the perceived cell pushing strength may be most affected by the sub-sampling strategy employed here, as even highly mixed patterns will begin to appear segregated when viewed with a higher magnification. Validation of our inference method (Fig 5f) would suggest that the majority of sub-regions across these samples were indeed consistent with weak cell pushing, however it is also plausible that stronger cell pushing was acting within these tumours, which led to the observed fragmented patterns of tumour cells, which in turn necessitated the sub-region analysis in these samples.

The observed distributions of inferred values for mutant replicative advantage, $s$, and cell pushing, $q$, within the same BaseScope image could be due to a number of reasons. The experimental tissue represents 2-dimensional samples taken from a 3-dimensional system along a randomly oriented plane. As previously discussed, this sampling can have significant impacts on the appearance of the sub-clonal pattern, potentially creating the false appearance of spatially distinct sub-clones. When analysed individually, these small isolated sub-clones are likely to lead to variances in the inferred dynamics. Beyond this, however, such variances in inferred parameter values could reflect local fluctuations in the dynamics of the system. Cell mixing is likely to be a result of a complex combination of cell-intrinsic and extrinsic factors, influenced by physical stresses exerted by the tumour microenvironment. Physical stresses may fluctuate even across small areas of tissue, and inferring fluctuating values of cell pushing in different areas of the tumour may be consistent with this notion. In line with this reasoning, the observed distribution of mutant replicative advantage, $s$, could reflect the varying propensity of mutant cells to proliferate, which can be expected to fluctuate spatially. Whilst certain genetic alterations may confer a cell-intrinsic increase in proliferation rate, other cell-extrinsic factors are also expected to contribute to the "effective" replication rate of any particular cell, such as the availability of free space, access to nutrients, and proximity to the immune system.

## Simulating sub-clonal growth in 3-dimensions

An important limitation of our agent-based model is that it simulates growing tumours explicitly in 2 spatial dimensions. We chose not to perform the majority of our simulations in 3D in order to reduce the complexity of our spatial data, allowing us to focus on developing the CMFPT analysis framework. It is likely, however, that tumour sub-clones will exhibit different topologies when simulated in 2D, compared to 2D samples of 3D tumours. The latter, of course, more closely resembles the BaseScope images to which we compare our simulated data, and so it is necessary to explore how our spatial analysis framework would perform when applied to 3D simulated tumours.

To explore this, we extended the same agent-based model from 2D to 3D, whilst preserving all other features and rules of the model other than the final system size, $N_{max}$, which we increased from $10^5$ to $10^7$ cells. To simulate the spatial sampling of real colorectal cancer tissue, we then extracted 2D slices of the simulated 3D tumours along three orthogonal axes. This yielded, for each simulated tumour, a set of 2D sub-clonal mixing patterns (S30(a) Fig).

Qualitatively, the patterns of WT and sub-clonal population mixing are far more diverse for sampled 3D tumours compared to 2D tumours (S30(b) Fig). This is especially true for boundary-driven growth tumours ($q = 0$). When simulated in 2D, boundary-driven growth gave rise to a low diversity set of sub-clone topologies, with strong segregation of WT and mutated populations, and mutated sub-clones which exhibited a distinct conical shape. Interestingly, when boundary-driven growth is simulated in 3D, the resulting 2D samples are far more diverse, both in terms of sub-clone shape and the degree of WT and mutant population mixing. These simulations reveal that, even in the absence of any cell pushing, population mixing is possible

depending on the orientation of 2D samples obtained from the 3D tumour, and suggest that some of the BaseScope samples to which we previously ascribed non-zero values of cell pushing, using 2D simulated patterns, could in fact be consistent with boundary-driven growth.

Moreover, we find that the ratio of WT to mutant cell populations varies significantly for different 2D samples of the same 3D tumour (S30(b) Fig). This will likely impact the perceived sub-clonal selection strength, $s$, and time of emergence, $n_{mut}$, and demonstrates that the relationship between these parameters and the resulting ratio of WT to mutant sub-clonal cells is far more complex in 3D versus 2D simulated tumours.

The difference between 2D and 3D derived mixing patterns is less pronounced for non-zero values of cell pushing, although a range of WT to mutant frequency ratios is also observed in this regime (S31 Fig). Interestingly, we note that the set of sub-clonal patterns generated under any given dynamics in 2D are also observed in the 3D sampled tumour, and represent a sub-set of the possible patterns under the corresponding 3D system. This is an important observation as, despite showing that the observed spatial patterns are different when simulations are extended from 2D to 3D, it provides some validation of our approach utilising 2D simulated patterns, showing that any given sub-clonal dynamics simulated in 2D results in a limited, but not unrealistic, representation of the same dynamics in the more complex 3D systems.

Analysis of a small number of sampled 3D simulated tumours using CMFPT further suggests that explicit 3D systems are far more complex than explicit 2D systems (S32 Fig). Parameter combinations which lead to narrow distributions of CMFPT measurements in 2D tumours often give rise to more diffuse and widely distributed data under the corresponding 3D system. This highlights the stark increase in complexity in 3D compared to 2D systems and the challenges of accounting for the effects of physical tissue sampling of 3D systems in real tumours. Whilst we have provided the first look at understanding the dynamics of a system using spatial mutation signatures, future work must address the complex tissue sampling process and better characterise its effect on the perceived dynamics.

## Discussion

The challenges involved with obtaining longitudinal samples in cancer studies calls for novel methods to analyse single time sampled data, especially spatial information, to unravel early tumour dynamics *in vivo*. Spatially resolved tumour samples have been studied in a multitude of ways in the prognosis of cancer, yet spatial information may also have the potential to inform on the past sub-clonal dynamics. Here, we leveraged statistics of random walkers as a new methodology to quantify complex patterns of sub-clonal mixing observed *in vivo*. We first characterised the normalised class mean first passage time (CMFPT) using a set of standard two-class artificial cell mixing patterns. By normalising each image measurement to that from a null model, in which the image pixels are rearranged uniformly at random, we were able to compare CMFPT data for patterns of different total sizes, and different ratios of the two classes. These simple measurements demonstrated the power of the CMFPT to distinguish between patterns of different class ratios and underlying pattern structures. Patterns with weak and strong segregation of the two classes were represented in radically different regions of the phase space.

Extending our analysis to a large dataset of simulated sub-clonal mixing patterns generated with our agent-based model, we showed that the CMFPT is capable of distinguishing between different regimes of sub-clonal dynamics, through the resulting patterns of sub-clonal mixing. Tumours with weakly advantageous mutant sub-clones, which emerged late during the WT population expansion, led to spatial patterns with vastly different first passage times to those

with more rapidly replicating mutant sub-clones, those with relatively early emerging sub-clones, or both. Consistent with other recent computational modelling based on genomic sequencing data [4, 33], we find that earlier arising mutant sub-clones lead to greater variegation in the final tumour. Whilst these effects have been studied through multiomic sequencing analysis of spatial bulk sub-samples, here we show for the first time that these dynamics can also be measured using the CMFPT applied to cellular resolution maps of the sub-clonal architecture.

These analyses demonstrate how the CMFPT provides a more detailed and informative description of the patterns of mutant sub-clonal mixing than the spatial Shannon's entropy measure that was previously used to analyse the BaseScope patterns. Whilst the agent-based model used in this study and the original BaseScope study were similar, here we also explored the influence of varying the strength of cell pushing. We showed that cell pushing can profoundly affect the resulting mutant sub-clonal pattern. Whilst the patterns produced under a surface growth model and one with cell pushing were qualitatively similar when analysed using Shannon's entropy, such dynamics were measurable when analysing the mixing patterns at multiple spatial scales using the CMFPT.

By applying our CMFPT method to human colorectal cancer samples analysed with Base-Scope, we compared statistical features of the experimental patterns to the simulated sub-clonal patterns. In doing so we estimated values for the replicative advantage of sub-clonal cells, $s$, the relative time at which the sub-clone infiltrated the local WT population, $n_{mut}$, and the strength of cell pushing, $q$. We inferred a wide range of dynamics, with some samples consistent with neutral sub-clonal evolution, and others suggesting strongly positive sub-clonal selection, with mutated cells replicating up to 4 times faster than WT cells (S2 Table). By analysing sub-regions of the sub-clonal patterns, we inferred local values for the relative time of sub-clonal emergence. In part due to confounding factors of inter-patient variability and stochasticity relating to the tissue sampling process, the range of predicted values was large, with some sub-regions consistent with very early sub-clonal populations that emerged after only 100 WT divisions ($n_{mut} = 0.001$) and others suggesting the WT population had undergone up to 50,000 divisions prior to the emergence of the sub-clonal population ($n_{mut} = 0.5$). In general, boundary driven growth ($q = 0$) or weak cell pushing ($q = 5$) was favoured over strong cell pushing ($q = 10$ & $q = 20$).

By analysing multiple sub-regions of the sub-clonal patterns in BaseScope samples 02, 28 and 34 we were able to obtain a more detailed insight into the sub-clonal dynamics in these tumours. When viewed in aggregate, the measurements within these samples formed a trend suggestive of highly aggressive mutant sub-clones, with rates of replication as high as 3 or 4 times that of the WT population. Whilst we were only able to infer local values for the relative time of sub-clonal invasion, the trend formed by these measurements allows us to speculate at the time of the original mutational event in the tumours. For instance, BaseScope sample 34 contains a greater fraction of sub-regions which are best described by very early invading sub-clones compared to samples 02 and 28. This suggests that the original mutational event occurred earlier in the evolution of this tumour than those in samples 02 and 28. To make a more qualitative statement would require extensive 3-dimensional modelling and remains a challenge for future work. However, our analysis of these samples demonstrates how important the role of sampling is in quantifying tumour evolution and how additional dynamics can be revealed by comparing sub-regions within the same tumour.

Many of the experimental sub-clonal patterns we analysed contained spatially disparate sub-clonal regions, yet it is rare for the same mutation to occur independently in multiple regions of any tumour. Despite some evidence of parallel evolution in clear-cell renal cell carcinoma (ccRCC) [34–36], where the same driver mutation occurs simultaneously in multiple

cells within the tumour, evidence of this phenomenon in colorectal cancer is limited. If this were to have occurred for the specific point mutations which were stained for using BaseScope, however, it may lead to multiple, potentially non-contiguous, regions of mutant cells. Since our computational model does not account for multiple occurrences of the same mutation, patterns of parallel evolution in our experimental samples would most likely be explained within our model framework by high cell pushing strength, $q$, since large values of this parameter can lead to the fragmentation of mutant cell clusters. Our computational model could be adapted straight-forwardly to model parallel evolution where, for example, mutations occur within two or more WT cells at similar times. Such a model could then be applied to study the spatial population mixing patterns in cases of ccRCC where parallel evolution has been observed using multi-region sequencing data, if data on the spatial mutational patterns were available.

Alternatively, the occurrence of spatially distinct sub-clonal regions could indicate modes of cell movement other than the passive cell pushing mechanic which we implement in our model. For instance, short range cell migration may also have important implications for sub-clonal mixing in colorectal cancers [7], especially in the early stages of tumourigenesis, and cell migration has been studied in other computational models of tumour growth [8, 37, 38]. We did not include these dynamics within our model, however future research should be devoted to understanding the spatial signatures of short range cell migration.

Alongside the measured sub-clonal patterns published in the original paper on BaseScope (Ref [20]) we also applied our method to a small number of colorectal cancer samples published in a recent study by Househam et al. [33]. In their work, Househam and colleagues combine multi-region sequencing data of colorectal tumours with spatial computational modelling to infer values for sub-clonal selection, relative time of sub-clone invasion and cell pushing strength. Samples from two patients in this study were also analysed with BaseScope (samples A7, A10 & A11 from patient C537, and A12 from patient C539; S10(a) Fig), giving us an opportunity to compare the modelling framework of Househam and colleagues to our spatial analysis approach as a means to measure the sub-clonal dynamics within these tumours. For instance, positive sub-clonal selection was predicted by the model of Househam et al. in patient C539. Analysing BaseScope sample A12 from the same patient using CMFPT, we also inferred positive sub-clonal selection (indicated by $s = 1$), and a late emerging sub-clone ($n_{mut} = 0.5$) which is consistent with the predictions made by Househam and colleagues.

Our predictions for patient C537, however, deviate from those made by Househam and colleagues. Our modelling suggested strong sub-clonal selection (best-fit values of $s = 3$ for each of samples A7, A10 & A11) which, whilst self-consistent, differ significantly from the predictions of Househam et al. who predicted neutral sub-clonal evolution in this tumour. It is not likely that these values of selection were erroneously ascribed due to biases in our inference approach (Fig 5f) suggesting that, within the confines of our agent-based model, these patterns were indeed consistent with large selection strengths. The disparity between our predictions for these samples and those made by Househam and colleagues highlight the challenges associated with the stochastic tissue sampling process, and the limited statistical power of an approach which utilises single spatial samples with low-depth genetic information. The multi-region whole genome sequencing performed by Househam et al. enabled them to employ more traditional statistical methods to measure sub-clonal selection [39, 40]. Furthermore, the spatial information derived from their sampling strategy assisted the reconstruction of ancestral relationships between cell lineages, which is a well-established and ostensibly more robust approach to detecting sub-clonal selection and placing bounds on sub-clone age [41–45]. Given that the tissue samples for patients C537 and C539 used in this study were different to those used by Househam et al., albeit from the same patients, and given the vastly different

techniques and quantity of data used in both studies, it is difficult to compare the two methods closely. It is likely, however, that the approach of Househam *et al.* offers a more accurate description of the sub-clonal dynamics than the spatial analysis used in this study. Nevertheless our work shows that, despite carrying little genetic information, the high spatial resolution BaseScope data encodes important information about the sub-clonal dynamics, and our CMFPT method can be used to summarise these high-dimensional measurements in a lower-dimensional space whilst preserving the important features of the patterns.

The most obvious simplification of the computational model is that it neglects the third spatial dimension. The BaseScope data are an example of 2D samples taken from 3D systems, and it is likely that comparing these to simulated patterns explicitly generated in 2D will impact our inferences on the sub-clonal dynamics. Despite this, our modelling approach is consistent with the majority of spatial modelling in the literature, including that of Househam *et al.*, which tends to be performed in 2D, mostly for common reasons of computational efficiency and model complexity. It may be, however, that the unique dependence of our method on the specific spatial arrangement of cells within the tumour makes our approach more sensitive to errors associated with neglecting the third spatial dimension. We sought to understand at a basic level the relationship between sub-clonal patterns sampled from 2D and 3D tumours with the same underlying sub-clonal dynamics. To do this, we extended our spatial simulations to 3D and simulated tumours for a small number of ($s$, $n_{mut}$, $q$) parameter combinations, extracting 2D slices of cells from these 3D systems in a process which more faithfully represented the tissue acquisition process in our experimental samples. Visually, the sub-clonal patterns obtained from 3D simulated tumours highlighted both that intermixing of WT and sub-clonal tumour populations is possible even without any cell pushing (*i.e.* $q = 0$) in 3D but not 2D simulated tumours, and that the coupling of population size and sub-clonal dynamics, observed in 2D simulated tumours, is markedly weaker in 2D samples obtained from 3D simulated tumours. This is in part why we did not include information on $\phi$ to infer sub-clonal dynamics using our 2D model, since doing so may have actually obscured our conclusions about the dynamics in the 3D experimental tumours. Future work must address and characterise the influence of tissue sampling to better exploit the information contained within the measured sub-clonal patterns.

A further limitation of our agent-based model is the absence of the surrounding non-cancerous tissue. We chose to exclude these cells in our spatial simulations for two main reasons. First, whilst physical forces exerted by the tumour microenvironment on cancer cells, and their impact on tumour architecture, have been studied [46–48], these effects are likely to suffer from significant inter-patient variability, and indeed there remains no consensus on how to model such interactions *in silico*. Whilst one could, in principle, perform simulations which included a third population, representing all other non-cancerous cells, and simulate random walks on the three-colour patterns, the increased complexity of such a model may obscure any inferences made on the experimental data. Second, explicit modelling of non-cancerous cells would in turn necessitate careful identification of tertiary structures within the tissue, such as colonic crypts, and other physical obstacles which might influence the spatial arrangement of the growing tumour. As a result, we opted not to include these factors in our spatial model, in order to optimise the balance between model complexity and instructiveness. Instead, accepting the model limitations, when applying our analysis framework to real tumour data, we endeavoured to mitigate the impact of the third colour in our experimental patterns by manually filling in un-highlighted areas of the BaseScope images, where appropriate, and separately analysing spatially distinct regions of tumour tissue. Future work should involve spatial models which explicitly model the non-tumour cell population, preserving this

information in the experimental samples and exploiting the extra information contained within.

Overall, we predicted a wide range of dynamics across our different patient samples. Our limited 3D simulations suggest that stochasticity introduced by the tissue sampling process itself could play a role in the range of observed dynamics, however intrinsic inter-patient variability may also be an important factor to explain these observations. Cancer cells exist as part of a highly complex biological and ecological system, and many factors can influence the evolution of a tumour *in vivo* including patient genetic background, lifestyle, and metabolic state, as well as the time of oncogenesis and precise location of the tumour within the body. Indeed, previous studies of cancer evolution have reported similar levels of variability across patient cohorts, including previous spatial analysis of the same colorectal cancer samples analysed in this study [20] and studies such as that of Househam *et al.* [33] which employ more traditional sequencing approaches. By better understanding factors such as the influence of sampling tissue from 3D tumours, we may be able to gain a deeper understanding of these other biological sources of variability.

In addition to their relevance to real biological systems, our focus on parameters $s$, $n_{mut}$ and $q$ was motivated by our need for a simple generative model with parameters which each might affect different aspects of the sub-clonal spatial pattern. Specifically, mutant replicative advantage, $s$, impacts the size and shape of the mutant sub-clone, sub-clone emergence time, $n_{mut}$, affects the size of the sub-clone, its position within the tumour and the extent of variegation of the WT and mutant pattern, and cell pushing strength, $q$, influences sub-clone cluster size and the extent of mixing between WT and mutant populations. These dynamics alone were not capable of recapitulating the full range of spatial mutation patterns observed in real human tumours, but rather we used this simple model to explore the spatial signatures of these dynamics, and demonstrated that our CMFPT framework is capable of recovering the underlying model parameters based solely on the resulting spatial patterns of population mixing. In spite of our specific choice of model design, many other models could conceivably have been adopted here, incorporating other dynamics relevant to expanding tumour populations. In addition to modelling mechanical interactions between tumour and non-tumour cells, we also could have modelled competition for resources between tumour cells as well as interactions between tumour and stroma other than competition for space. Interactions between tumour cells and cancer associated fibroblasts have been associated with poor prognosis in colorectal cancer patients [49]. Similarly, interactions between tumour and immune cells are known to be an important process in the evolution of cancers, and specifically colorectal cancer [15–18]. The spatial distribution of tumour and non-tumour cells such as cancer associated fibroblasts and immune cells could influence the spatial patterns of the WT and sub-clonal tumour populations themselves, perhaps by inducing non-homogeneous tumour cell killing or growth, and such dynamics could be incorporated into simulations. We could not obtain information from our experimental samples on these stroma populations, however our CMFPT framework could potentially be used to study the spatial signatures of these interactions.

Dynamics related to non-genetic evolution, such as phenotypic switching, could also be incorporated into future simulation models, with the forward and backward switching rates likely impacting the observed spatial patterns. The pushing algorithm in our model could have been replaced by other similar dynamics, such as a local dispersal model in which dividing cells can place one daughter cell in a nearby empty lattice point, without pushing any cells in between. These dynamics could lead to similar spatial patterns to those produced with our pushing model, though perhaps with some difference in sub-clonal cluster size due to the reduction in the number of displaced cells per division event. We would expect that our

CMFPT framework could be used to measure the spatial signatures in a model with this type of local dispersal, relating the sub-clonal patterns to parameters such as dispersal rate or range.

In this study we explored the efficacy of the mean of the class first passage time to quantify spatial signatures of sub-clonal dynamics. In its current form, we showed that the CMFPT is similar to the mean shortest distance. Future implementations of our metric, however, could potentially go further than mean shortest distance by leveraging higher order moments of the first passage time distributions, such as combining information about the mean and variance for each of the 2-class transitions. In doing so, first passage time methods could possibly be used to quantify the neighbourhood of each cell, capturing more nuanced differences between different sub-clonal patterns, and in turn enabling more sophisticated and biologically relevant models to be fitted to the spatial patterns. Further research should be directed at exploiting the distribution of first passage time values to obtain a fuller description of the underlying sub-clonal dynamics.

In summary, we have developed the use of CMFPT as a new method of quantifying clustering and heterogeneity of sub-clonal patterns in spatially resolved tumour samples. We extended this method from previous applications in spatial demographic data to patterns of sub-clonal mixing in colorectal tumour samples acquired with BaseScope, and combined experimental measurements with spatial computational modelling of expanding tumour populations to estimate parameters of early sub-clonal evolution in our samples. In line with some contemporary models of tumour evolution [1, 4, 7], our analysis suggests that the large tumour sub-clones which were observed may have arisen early in the expansion of the WT tumour cell population, exhibiting a range of selective growth advantages over WT cells. Our work demonstrates the capability of CMFPT as a means to measure underlying dynamic parameters using their resulting spatial signatures, and the potential to understand early cancer evolution by applying CMFPT to high resolution spatial data of tumour sub-population boundaries. Further research to determine the influence of 2-dimensional sampling on inference of the dynamics, and effectively modelling the role of the tumour microenvironment, is required to further elucidate early cancer dynamics.

## Methods

### Ethics declaration

Formalin-fixed paraffin embedded (FFPE) tissue blocks were obtained from University College and St Mark's Hospitals, London, under multi-centre ethical approval (11/LO/1613); the QUASAR2 trial (ISRCTN45133151, ethical approval REC 09/H0606/5+5); the COIN trial (ISRCTN; 27286448); and the FOCUS trial (ISRCTN; 79877428, ethical approval 15/EE/0241). Written informed consent was waived by the relevant RECs due to the retrospective and anonymous nature of this study.

### Simulating random walkers and estimating first passage time statistics

To assess the spatial heterogeneity of cell types we use the normalised class mean first passage times $\tilde{\tau}_{\alpha\beta}$ between any pair of classes $\alpha$ and $\beta$ [25]. To estimate this quantity, we first compute the unnormalised class mean first passage time from class $\alpha$ to $\beta$ by simulating 5,000 random walks, and averaging over the resulting first passage times, for each starting node of class $\alpha$ in the image. To compute the class mean first passage time, we then average over this value across all $\alpha$ nodes. The normalised class mean first passage time, $\tilde{\tau}_{\alpha\beta}$, is obtained by dividing the unnormalised class mean first passage time by its null-model counterpart, $\tau_{\alpha\beta}^{\text{null}}$. To estimate $\tau_{\alpha\beta}^{\text{null}}$ we simulate random walks on a graph with the same connectivity as the original

graph, but with the node classes redistributed uniformly at random, such that for an original graph with class ratio $\phi = x$, any arbitrarily selected coloured node will be of class $\alpha$ and $\beta$ with probabilities $P_\alpha = x$ and $P_\beta = 1 - x$ respectively. For each pattern, the null model values $\tau_{\alpha\beta}^{\text{null}}$ are obtained over multiple realisations of this colouring process. Since the quantity $\tilde{\tau}_{\alpha\beta}$ is normalised by a null-model, it is dimensionless and represents whether the time to arrive at a node of a given class is smaller or larger than the corresponding transition in the null-model. For example, if $\tilde{\tau}_{\alpha\beta} < 1$ then the expected number of steps taken before arriving at a node of class $\beta$ is less in the experimental pattern than in the null-model, and is greater when $\tilde{\tau}_{\alpha\beta} > 1$.

To reduce computational time, we downsample the BaseScope data by binning image pixels. When altering the image resolution, the proportion of pixel classes is preserved in each bin. For example, if the pixels falling into a particular bin consisted of 50% red and 50% yellow pixels, then a walker on the downsampled network will encounter either a red or yellow bin at these coordinates with equal probabilities of $P_{red} = P_{yellow} = \frac{1}{2}$. We downsample all BaseScope and simulated sub-clonal patterns to the same resolution, 3000 pixels in total, prior to analysis.

## Spatial simulations of sub-clonal evolution in tumours

We generate cell mixing patterns using an algorithm based on the Gillespie algorithm [50] to simulate stochastic birth and death of cells on a 2-dimensional square lattice. Since the Base-Scope assay targets and detects specific genetic point mutations, we adopt a binary system in which a particular cell can either be wild-type (WT) or mutated. A simulation begins by seeding a single WT cell on the lattice, which has a birth rate greater than its death rate, and terminates when the entire system (WT and mutated cells) reaches a specified size $N_{max}$. During initial growth, the WT clone expands until it reaches a size $n = N_{max} \cdot n_{mut}$ cells, at which point an existing cell is randomly chosen to acquire a mutation. All subsequent cells related to the newly mutated cell will inherit the mutation, and cells can not revert from the mutated state back to WT.

Cells carrying the mutation experience a relative fitness advantage compared to WT cells. This fitness advantage manifests as an increased birth rate, giving the birth rate of a mutated cell, $b_{mut}$, as

$$b_{mut} = (1 + s) \cdot b_{WT}, \qquad (2)$$

where $s$ represents the relative fitness advantage conferred by mutations and $b_{WT}$ is the replication rate of WT cells. Our model assumes that cell cycle times are Poissonian distributed. Whilst in reality cell cycle distributions are more tightly peaked, this assumption has been shown to reduce lattice artefacts in cellular automaton models [51]. Cell death is coupled to cell birth in the simulations. Before a cell divides, it is randomly determined either to continue with the division, or die giving a death rate for cell $i$, $d_i$, of

$$d_i = b_i \cdot \psi, \qquad (3)$$

where $\psi \in [0, 1]$, and is fixed at $\psi = 0.3$ throughout this study unless stated otherwise. This implies that cells which divide at a faster rate also die at a faster rate, which could reflect the greater rate of apoptosis resulting from faster cell cycling in these cells. When a cell dies it immediately is removed from the lattice leaving behind an empty lattice point.

To model the displacement of neighbouring cells during division, we adapted an algorithm developed by Waclaw et al. [8], which involves the dividing cell searching its neighbourhood and finding a "path" to one of the nearest empty lattice points. In order to create space to

divide into two daughter cells, the dividing cell pushes neighbouring cells along this path, filling the nearby empty lattice point and creating a new empty space adjacent to itself. In our simulations we implement these mechanics and extend them by accounting for relative positions of cells during pushing, favouring straight-line pushing over displacement in other directions, in accordance with Newton's second law of motion. The parameter $q$ determines the maximum allowed size for the constructed path in this algorithm. If no path can be found which is less than or equal to $q$ in length, the cell cannot successfully divide. Setting this parameter to $q = 0$ leads to surface growth dynamics, in which a cell must be adjacent to an empty lattice point in order to divide successfully.

The range of values used for mutant selective advantage, $s \in \{0, 0.1, 0.2, 0.5, 1, 2, 3\}$, was chosen so that we were able to investigate spatial signatures of neutral sub-clonal growth, through intermediate positive selection, up to very strong positive selection. These values span the expected range of selective coefficients related to driver point mutations [52, 53]. Our chosen range of $n_{mut}$ was naturally restricted between $0 < n_{mut} < 1$. We chose to explore values spanning several orders of magnitude ranging $n_{mut} \in \{0.001, 0.01, 0.03, 0.05, 0.08, 0.1, 0.5\}$ leading to, for $N_{max} = 1 \times 10^5$, the earliest emerging mutant populations appearing after just 100 WT divisions ($n_{mut} = 0.001$) and the latest appearing after 50,000 WT divisions ($n_{mut} = 0.5$). Our motivation for our chosen range of pushing values, $q \in \{0, 5, 10, 20\}$, was to simulate a spectrum of pushing dynamics ranging from boundary driven growth ($q = 0$) up to increasingly stronger pushing strengths approaching the volumetric (exponential) growth regime. Whilst our implementation of cell pushing is somewhat abstract, and one could conceive of other approaches to modelling cell movement on the lattice, the mode of growth in human tumours and its relation to tumour type and stage is not well known. This model design allows us to explore the spatial signatures of the different regimes and smoothly transition between different modes of growth using the parameter $q$.

Model parameters used to generate *in silico* mixing patterns are listed in S1 Table. Note that, due to decreased probability of sub-clonal survival for $(s, n_{mut}, q) = (0, 0.5, 20)$, $(0.1, 0.5, 20)$ and $(0.2, 0.5, 20)$, sub-clonal pattern data could not be generated for these parameter combinations.

## Bayesian grid-search for best-fit model parameter inference

To perform inference of mutant replicative advantage ($s$), mutation timing ($n_{mut}$) and cell pushing strength ($q$) for each BaseScope sub-sampled region (S12–S27 Figs), we implement a Bayesian-style grid search. For each experimental pattern, we construct the posterior distribution of $s$, $n_{mut}$ and $q$ by finding the $n$ nearest simulated sub-clonal patterns in the 4-dimensional phase space of class mean first passage times $\tilde{\tau}_{rr}$, $\tilde{\tau}_{ry}$, $\tilde{\tau}_{yr}$ and $\tilde{\tau}_{yy}$, as determined by log-Euclidean distance. We chose a sample size of $n = 100$ for each experimental pattern, and computed 95% credible regions for our estimations of each parameter value. To select the "best fit" simulated sub-clonal pattern we found the most abundant combination of $s$, $n_{mut}$ and $q$ within the posterior sample set and assigned these parameter values as point estimates for the experimental pattern. Information on the mutant frequency $\phi$, could be used to supplement the CMFPT measurements during our grid-search inference, however this ratio is highly susceptible to uncertainty introduced by the tissue sampling process, as evidenced by our exploratory 3D simulations which demonstrate that measured $\phi$ can vary wildly depending on the orientation of the 2D sample acquired from the 3D tumour. Measurements of $\phi$ could in principle be incorporated into the grid-search analysis, however in this study we were interested in exploring the efficacy of CMFPT measurements alone in quantifying sub-clonal dynamics based on the appearance of the mutant sub-clones.

## Shannon's entropy calculations

We replicated the calculations of Shannon's entropy from Ref [20] and applied these to our own set of simple 2-dimensional mixing patterns generated using the clusters, centred and column models (S1 Fig). To calculate the Shannon's entropy of a pattern, we divide the pattern into $n$ non-overlapping quadrats with dimensions $L \times L$ cells, and compute the frequency of mutant cells within each quadrat. We then compute Shannon's entropy, $H$, as

$$H = -\frac{1}{n} \sum_{i=1}^{n} \left[ p_i \log_2 p_i + (1 - p_i)\log_2(1 - p_i) \right], \qquad (4)$$

where $p_i$ denotes the frequency of mutant cells in the $i^{\text{th}}$ quadrat. When analysing our 2-dimensional mixing patterns, which themselves have dimensions of $54 \times 54$ cells, we use quadrats with dimensions $10 \times 10$. As has been discussed with respect to quadrat-based methods for estimating fractal dimension [54–56], the estimation of Shannon's entropy using this method can change depending on the placement of the grid in the $(x, y)$ plane. To address this, we compute $H$ for each pattern for a number of $x$ and $y$ grid offsets, specifically from 0 up to $L$ in both orthogonal directions independently. We then take the minimum measured value of $H$ across all $x$ and $y$ offsets.

## Mean shortest distance calculations

We compute the mean shortest distance [57] on our simple 2-dimensional mixing patterns generated using the clusters, centred and column models (S3 Fig). As with our CMFPT calculations, we obtain four quantities from our 2-colour patterns, $\tilde{d}_{yy}$, $\tilde{d}_{yr}$, $\tilde{d}_{ry}$ and $\tilde{d}_{rr}$. For example, to compute the mean shortest distance from yellow to red, $\tilde{d}_{yr}$, we compute the average shortest path length in the adjacency network between cells (in terms of number of edges) between the pair of nodes $i$ and $j$ for $i \in Y$ and $j \in R$

$$\langle d_{yr} \rangle = \frac{1}{|Y| \times |R|} \sum_{\substack{i \in Y \\ j \in R}} d_{ij}, \qquad (5)$$

where $Y$ and $R$ denote the set of all yellow and red nodes, respectively, and $d_{ij}$ denotes the shortest path length connecting cells $i$ and $j$. To obtain $\tilde{d}_{yr}$, $\langle d_{yr} \rangle$ is normalized by $\langle d_{yr}^{\text{null}} \rangle$, which is the same quantity calculated over a null-model where colors are reshuffled at random.

## Supporting information

**S1 Fig. Characterising 2-dimensional mixing patterns using Shannon's entropy.** Shannon's entropy analysis using the approach described in Ref [20]. Validation of our algorithm using **(a)** fully segregated and **(b)** fully mixed patterns, where the latter value represents mean ± standard deviation for entropy values of 10 randomly generated fully mixed patterns ($\phi = 0.5$ in both cases). **(c, d)** Shannon's entropy of patterns generated using the clusters; **(e, f)** centred and **(g, h)** column models. **(i)** Location of the patterns obtained for the three models with varying class ratio, $\phi$. All patterns have dimensions of $54 \times 54$, and Shannon's entropy was computed using quadrats of size $10 \times 10$.
(TIF)

**S2 Fig. Characterising 2-dimensional simulated sub-clonal patterns using Shannon's entropy.** Shannon's entropy analysis of the same set of simulated tumour sub-clonal patterns as were analysed using CMFPT. Points are coloured according to values of model parameter $s$

and point shape depends on parameter $n_{mut}$. Data within each panel represent simulations for all possible combinations of $s$ and $n_{mut}$ where $s \in \{0, 0.1, 0.2, 0.5, 1, 2, 3\}$ and $n_{mut} \in \{0.001, 0.01, 0.03, 0.05, 0.08, 0.1, 0.5\}$ (approximately 100 simulated patterns for each parameter combination). Images are separated depending on their pushing value $q = 0$ (**a**), $q = 5$ (**b**), $q = 10$ (**c**) and $q = 20$ (**d**). Inset within each panel is a magnified section of the phase space spanning approximately $0 < \phi < 0.1$ and $0 \leq$ Shannon's entropy $< 0.1$.
(TIF)

**S3 Fig. Characterising 2-dimensional mixing patterns using mean shortest distance.** Normalised mean shortest distance analysis of patterns generated using the **(a, b)** clusters; **(c, d)** centred and **(e, f)** column models. **(g)** Location of the patterns obtained for the three models with varying class ratio, $\phi$. Normalised mean shortest distance from red to yellow cells is denoted $\tilde{d}_{ry}$, and from yellow to red denoted $\tilde{d}_{yr}$.
(TIF)

**S4 Fig. Characterising 2-dimensional simulated sub-clonal patterns using mean shortest distance.** Normalised mean shortest distance analysis of the same set of simulated tumour sub-clonal patterns as were analysed using CMFPT. Points are coloured according to values of model parameter $s$ and point shape depends on parameter $n_{mut}$. Data within each panel represent simulations for all possible combinations of $s$ and $n_{mut}$ where $s \in \{0, 0.1, 0.2, 0.5, 1, 2, 3\}$ and $n_{mut} \in \{0.001, 0.01, 0.05, 0.1, 0.5\}$ (approximately 100 simulated patterns for each parameter combination). Images are separated depending on their pushing value $q = 0$ (**a**), $q = 5$ (**b**), $q = 10$ (**c**) and $q = 20$ (**d**).
(TIF)

**S5 Fig. Analysis of simulated sub-clonal patterns. (a-d)** Representative examples of tumour sub-clonal patterns simulated with a pushing strength of $q = 0$. **(e)** CMFPT analysis of all simulated sub-clonal patterns with a pushing strength of $q = 0$ in the $(\tilde{\tau}_{ry}, \tilde{\tau}_{ry}/\tilde{\tau}_{yr})$ phase space. Data represent simulations for all possible combinations of $s$ and $n_{mut}$ where $s \in \{0, 0.1, 0.2, 0.5, 1, 2, 3\}$ and $n_{mut} \in \{0.001, 0.01, 0.03, 0.05, 0.08, 0.1, 0.5\}$, with $q = 0$ (approximately 100 simulated patterns for each parameter combination). Points are coloured according to pattern class ratio, $\phi$, and images shown in (a-d) are highlighted in the phase space.
(TIF)

**S6 Fig. Analysis of simulated sub-clonal patterns. (a-d)** Representative examples of tumour sub-clonal patterns simulated with a pushing strength of $q = 10$. **(e)** CMFPT analysis of all simulated sub-clonal patterns with a pushing strength of $q = 10$ in the $(\tilde{\tau}_{ry}, \tilde{\tau}_{ry}/\tilde{\tau}_{yr})$ phase space. Data represent simulations for all possible combinations of $s$ and $n_{mut}$ where $s \in \{0, 0.1, 0.2, 0.5, 1, 2, 3\}$ and $n_{mut} \in \{0.001, 0.01, 0.03, 0.05, 0.08, 0.1, 0.5\}$, with $q = 10$ (approximately 100 simulated patterns for each parameter combination). Points are coloured according to pattern class ratio, $\phi$, and images shown in (a-d) are highlighted in the phase space.
(TIF)

**S7 Fig. Analysis of simulated sub-clonal patterns. (a-d)** Representative examples of tumour sub-clonal patterns simulated with a pushing strength of $q = 20$. **(e)** CMFPT analysis of all simulated sub-clonal patterns with a pushing strength of $q = 20$ in the $(\tilde{\tau}_{ry}, \tilde{\tau}_{ry}/\tilde{\tau}_{yr})$ phase space. Data represent simulations for all possible combinations of $s$ and $n_{mut}$ where $s \in \{0, 0.1, 0.2, 0.5, 1, 2, 3\}$ and $n_{mut} \in \{0.001, 0.01, 0.03, 0.05, 0.08, 0.1, 0.5\}$, with $q = 20$ (approximately 100 simulated patterns for each parameter combination). Points are coloured according to pattern class ratio, $\phi$, and images shown in (a-d) are highlighted in the phase space.
(TIF)

**S8 Fig. Analysis of simulated sub-clonal patterns for varying model parameter values.**
Images generated by the model in the phase space $(\tilde{\tau}_{ry}, \tilde{\tau}_{ry}/\tilde{\tau}_{yr})$, with points coloured according to value of $\phi$. Data within each panel represent simulations for all possible combinations of $s$ and $n_{mut}$ where $s \in \{0, 0.1, 0.2, 0.5, 1, 2, 3\}$ and $n_{mut} \in \{0.001, 0.01, 0.03, 0.05, 0.08, 0.1, 0.5\}$ (approximately 100 simulated patterns for each parameter combination). Images are separated depending on their pushing value **(a)** 0, **(b)** 5, **(c)** 10 and **(d)** 20.
(TIF)

**S9 Fig. Mean and variance of CMFPT in simulated sub-clonal patterns.** Representative data showing distribution (mean and variance) of CMFPT measured in simulated sub-clonal patterns for various $s$, $n_{mut}$ and $q$ combinations. For each combination of parameters, we simulated approximately 100 tumours and quantified the sub-clonal patterns using CMFPT. Top row: data for **(a)** $n_{mut} = 0.001$ and **(b)** $n_{mut} = 0.5$ tumour, both with $s \in \{0, 0.1, 0.2, 0.5, 1, 2, 3\}$. Bottom row: data for **(c)** $s = 0$ and **(d)** $s = 3$ tumours, both with $n_{mut} \in \{0.001, 0.01, 0.03, 0.05, 0.08, 0.1, 0.5\}$. Data are mean ± s.d.
(TIF)

**S10 Fig. All BaseScope patterns analysed in this study. (a)** Sub-clonal mutations in colorectal tumours analysed with BaseScope used in this study, from Refs [20, 33]. Colorectal tumour cells and surrounding epithelium is pictured, with wild-type tumour cells highlighted in yellow, and mutated sub-clonal tumour population highlighted in red. Non-cancerous tissue is not highlighted. Scale bars, where present, represent $2000\mu m$ **(b)** Representative example of a "raw" BaseScope image, and the same image after application of the pre-processing steps described in the main text.
(TIF)

**S11 Fig. Initial CMFPT analysis of unprocessed sub-clonal patterns in human colorectal tumour samples.** CMFPT analysis of human colorectal tumour samples plotted in the $(\tilde{\tau}_{ry}, \tilde{\tau}_{ry}/\tilde{\tau}_{yr})$ phase space. Each triangular marker represents a single colorectal cancer sample analysed with BaseScope. Circular points represent all simulated sub-clonal patterns, and points are coloured according to pattern class ratio, $\phi$.
(TIF)

**S12 Fig. Bayesian grid-search analysis of BaseScope sample 02. (a)** Sub-sample of sample 02. **(b)** Marginal posterior distributions of model parameters $s$, $n_{mut}$, and $q$, representing mutant selection strength, mutation timing and cell pushing strength respectively. Inferred parameter value is indicated by the vertical dashed line along the diagonal panels. 95% credible regions lie within the shaded region in the diagonal panels. Where no shaded region is given, this interval was the entire parameter range. **(c)** All analysed simulated sub-clonal mixing patterns (grey points) with CMFPT value of the BaseScope sub-sample (star) and posterior samples (green points). **(d)** Best-fit simulated sub-clonal pattern and parameters representing the most abundant parameter combination within the posterior distribution.
(TIF)

**S13 Fig. Bayesian grid-search analysis of BaseScope sample 11. (a)** Sub-sample of sample 11. **(b)** Marginal posterior distributions of model parameters $s$, $n_{mut}$, and $q$, representing mutant selection strength, mutation timing and cell pushing strength respectively. Inferred parameter value is indicated by the vertical dashed line along the diagonal panels. 95% credible regions lie within the shaded region in the diagonal panels. Where no shaded region is given, this interval was the entire parameter range. **(c)** All analysed simulated sub-clonal mixing patterns (grey points) with CMFPT value of the BaseScope sub-sample (star) and posterior samples (green

points). **(d)** Best-fit simulated sub-clonal pattern and parameters representing the most abundant parameter combination within the posterior distribution.
(TIF)

**S14 Fig. Bayesian grid-search analysis of BaseScope sample 12. (a)** Sub-sample of sample 12. **(b)** Marginal posterior distributions of model parameters $s$, $n_{mut}$, and $q$, representing mutant selection strength, mutation timing and cell pushing strength respectively. Inferred parameter value is indicated by the vertical dashed line along the diagonal panels. 95% credible regions lie within the shaded region in the diagonal panels. Where no shaded region is given, this interval was the entire parameter range. **(c)** All analysed simulated sub-clonal mixing patterns (grey points) with CMFPT value of the BaseScope sub-sample (star) and posterior samples (green points). **(d)** Best-fit simulated sub-clonal pattern and parameters representing the most abundant parameter combination within the posterior distribution.
(TIF)

**S15 Fig. Bayesian grid-search analysis of BaseScope sample 13. (a)** Sub-sample of sample 13. **(b)** Marginal posterior distributions of model parameters $s$, $n_{mut}$, and $q$, representing mutant selection strength, mutation timing and cell pushing strength respectively. Inferred parameter value is indicated by the vertical dashed line along the diagonal panels. 95% credible regions lie within the shaded region in the diagonal panels. Where no shaded region is given, this interval was the entire parameter range. **(c)** All analysed simulated sub-clonal mixing patterns (grey points) with CMFPT value of the BaseScope sub-sample (star) and posterior samples (green points). **(d)** Best-fit simulated sub-clonal pattern and parameters representing the most abundant parameter combination within the posterior distribution.
(TIF)

**S16 Fig. Bayesian grid-search analysis of BaseScope sample 16. (a)** Sub-sample of sample 16. **(b)** Marginal posterior distributions of model parameters $s$, $n_{mut}$, and $q$, representing mutant selection strength, mutation timing and cell pushing strength respectively. Inferred parameter value is indicated by the vertical dashed line along the diagonal panels. 95% credible regions lie within the shaded region in the diagonal panels. Where no shaded region is given, this interval was the entire parameter range. **(c)** All analysed simulated sub-clonal mixing patterns (grey points) with CMFPT value of the BaseScope sub-sample (star) and posterior samples (green points). **(d)** Best-fit simulated sub-clonal pattern and parameters representing the most abundant parameter combination within the posterior distribution.
(TIF)

**S17 Fig. Bayesian grid-search analysis of BaseScope sample 17a1. (a)** Sub-sample of sample 17a1. **(b)** Marginal posterior distributions of model parameters $s$, $n_{mut}$, and $q$, representing mutant selection strength, mutation timing and cell pushing strength respectively. Inferred parameter value is indicated by the vertical dashed line along the diagonal panels. 95% credible regions lie within the shaded region in the diagonal panels. Where no shaded region is given, this interval was the entire parameter range. **(c)** All analysed simulated sub-clonal mixing patterns (grey points) with CMFPT value of the BaseScope sub-sample (star) and posterior samples (green points). **(d)** Best-fit simulated sub-clonal pattern and parameters representing the most abundant parameter combination within the posterior distribution.
(TIF)

**S18 Fig. Bayesian grid-search analysis of BaseScope sample 17a2. (a)** Sub-sample of sample 17a2. **(b)** Marginal posterior distributions of model parameters $s$, $n_{mut}$, and $q$, representing mutant selection strength, mutation timing and cell pushing strength respectively. Inferred

parameter value is indicated by the vertical dashed line along the diagonal panels. 95% credible regions lie within the shaded region in the diagonal panels. Where no shaded region is given, this interval was the entire parameter range. **(c)** All analysed simulated sub-clonal mixing patterns (grey points) with CMFPT value of the BaseScope sub-sample (star) and posterior samples (green points). **(d)** Best-fit simulated sub-clonal pattern and parameters representing the most abundant parameter combination within the posterior distribution.
(TIF)

**S19 Fig. Bayesian grid-search analysis of BaseScope sample 17b. (a)** Sub-sample of sample 17b. **(b)** Marginal posterior distributions of model parameters $s$, $n_{mut}$, and $q$, representing mutant selection strength, mutation timing and cell pushing strength respectively. Inferred parameter value is indicated by the vertical dashed line along the diagonal panels. 95% credible regions lie within the shaded region in the diagonal panels. Where no shaded region is given, this interval was the entire parameter range. **(c)** All analysed simulated sub-clonal mixing patterns (grey points) with CMFPT value of the BaseScope sub-sample (star) and posterior samples (green points). **(d)** Best-fit simulated sub-clonal pattern and parameters representing the most abundant parameter combination within the posterior distribution.
(TIF)

**S20 Fig. Bayesian grid-search analysis of BaseScope sample 25. (a)** Sub-sample of sample 25. **(b)** Marginal posterior distributions of model parameters $s$, $n_{mut}$, and $q$, representing mutant selection strength, mutation timing and cell pushing strength respectively. Inferred parameter value is indicated by the vertical dashed line along the diagonal panels. 95% credible regions lie within the shaded region in the diagonal panels. Where no shaded region is given, this interval was the entire parameter range. **(c)** All analysed simulated sub-clonal mixing patterns (grey points) with CMFPT value of the BaseScope sub-sample (star) and posterior samples (green points). **(d)** Best-fit simulated sub-clonal pattern and parameters representing the most abundant parameter combination within the posterior distribution.
(TIF)

**S21 Fig. Bayesian grid-search analysis of BaseScope sample 28. (a)** Sub-sample of sample 28. **(b)** Marginal posterior distributions of model parameters $s$, $n_{mut}$, and $q$, representing mutant selection strength, mutation timing and cell pushing strength respectively. Inferred parameter value is indicated by the vertical dashed line along the diagonal panels. 95% credible regions lie within the shaded region in the diagonal panels. Where no shaded region is given, this interval was the entire parameter range. **(c)** All analysed simulated sub-clonal mixing patterns (grey points) with CMFPT value of the BaseScope sub-sample (star) and posterior samples (green points). **(d)** Best-fit simulated sub-clonal pattern and parameters representing the most abundant parameter combination within the posterior distribution.
(TIF)

**S22 Fig. Bayesian grid-search analysis of BaseScope sample 28 (continued). (a)** Sub-sample of sample 28. **(b)** Marginal posterior distributions of model parameters $s$, $n_{mut}$, and $q$, representing mutant selection strength, mutation timing and cell pushing strength respectively. Inferred parameter value is indicated by the vertical dashed line along the diagonal panels. 95% credible regions lie within the shaded region in the diagonal panels. Where no shaded region is given, this interval was the entire parameter range. **(c)** All analysed simulated sub-clonal mixing patterns (grey points) with CMFPT value of the BaseScope sub-sample (star) and posterior samples (green points). **(d)** Best-fit simulated sub-clonal pattern and

parameters representing the most abundant parameter combination within the posterior distribution.
(TIF)

**S23 Fig. Bayesian grid-search analysis of BaseScope sample 34. (a)** Sub-sample of sample 34. **(b)** Marginal posterior distributions of model parameters $s$, $n_{mut}$, and $q$, representing mutant selection strength, mutation timing and cell pushing strength respectively. Inferred parameter value is indicated by the vertical dashed line along the diagonal panels. 95% credible regions lie within the shaded region in the diagonal panels. Where no shaded region is given, this interval was the entire parameter range. **(c)** All analysed simulated sub-clonal mixing patterns (grey points) with CMFPT value of the BaseScope sub-sample (star) and posterior samples (green points). **(d)** Best-fit simulated sub-clonal pattern and parameters representing the most abundant parameter combination within the posterior distribution.
(TIF)

**S24 Fig. Bayesian grid-search analysis of BaseScope sample A7. (a)** Sub-sample of sample A7. **(b)** Marginal posterior distributions of model parameters $s$, $n_{mut}$, and $q$, representing mutant selection strength, mutation timing and cell pushing strength respectively. Inferred parameter value is indicated by the vertical dashed line along the diagonal panels. 95% credible regions lie within the shaded region in the diagonal panels. Where no shaded region is given, this interval was the entire parameter range. **(c)** All analysed simulated sub-clonal mixing patterns (grey points) with CMFPT value of the BaseScope sub-sample (star) and posterior samples (green points). **(d)** Best-fit simulated sub-clonal pattern and parameters representing the most abundant parameter combination within the posterior distribution.
(TIF)

**S25 Fig. Bayesian grid-search analysis of BaseScope sample A10. (a)** Sub-sample of sample A10. **(b)** Marginal posterior distributions of model parameters $s$, $n_{mut}$, and $q$, representing mutant selection strength, mutation timing and cell pushing strength respectively. Inferred parameter value is indicated by the vertical dashed line along the diagonal panels. 95% credible regions lie within the shaded region in the diagonal panels. Where no shaded region is given, this interval was the entire parameter range. **(c)** All analysed simulated sub-clonal mixing patterns (grey points) with CMFPT value of the BaseScope sub-sample (star) and posterior samples (green points). **(d)** Best-fit simulated sub-clonal pattern and parameters representing the most abundant parameter combination within the posterior distribution.
(TIF)

**S26 Fig. Bayesian grid-search analysis of BaseScope sample A11. (a)** Sub-sample of sample A11. **(b)** Marginal posterior distributions of model parameters $s$, $n_{mut}$, and $q$, representing mutant selection strength, mutation timing and cell pushing strength respectively. Inferred parameter value is indicated by the vertical dashed line along the diagonal panels. 95% credible regions lie within the shaded region in the diagonal panels. Where no shaded region is given, this interval was the entire parameter range. **(c)** All analysed simulated sub-clonal mixing patterns (grey points) with CMFPT value of the BaseScope sub-sample (star) and posterior samples (green points). **(d)** Best-fit simulated sub-clonal pattern and parameters representing the most abundant parameter combination within the posterior distribution.
(TIF)

**S27 Fig. Bayesian grid-search analysis of BaseScope sample A12. (a)** Sub-sample of sample A12. **(b)** Marginal posterior distributions of model parameters $s$, $n_{mut}$, and $q$, representing mutant selection strength, mutation timing and cell pushing strength respectively. Inferred

parameter value is indicated by the vertical dashed line along the diagonal panels. 95% credible regions lie within the shaded region in the diagonal panels. Where no shaded region is given, this interval was the entire parameter range. **(c)** All analysed simulated sub-clonal mixing patterns (grey points) with CMFPT value of the BaseScope sub-sample (star) and posterior samples (green points). **(d)** Best-fit simulated sub-clonal pattern and parameters representing the most abundant parameter combination within the posterior distribution.
(TIF)

**S28 Fig. Sub-section analysis of BaseScope sample 02. (a)** Best-fit simulated sub-clonal pattern shown next to each sub-section with corresponding model parameters. **(b)** Marginal distributions of inferred model parameters across all sub-sections.
(TIF)

**S29 Fig. Sub-section analysis of BaseScope sample 28. (a)** Best-fit simulated sub-clonal pattern shown next to each sub-section with corresponding model parameters. **(b)** Marginal distributions of inferred model parameters across all sub-sections.
(TIF)

**S30 Fig. Overview of 3-dimensional simulations. (a)** 3D simulated tumour with $(s, n_{mut}, q) = (1, 0.01, 0)$. 2D slices are extracted from 3D tumour, with the observed 2D spatial patterns analysed using fractal analysis. In total, approximately 10 3D tumours were simulated for each of the following combination of parameters; $(s, n_{mut}, q) = (0, 0.001, 0); (0, 0.1, 0); (0.5, 0.001, 0); (0.5, 0.5, 0); (1, 0.001, 0); (1, 0.1, 0); (1, 0.1, 10); (3, 0.001, 0); (3, 0.1, 0); (3, 0.1, 20); (3, 0.5, 0)$ **(b)** Examples of 2D slices obtained from the 3D tumour in (a).
(TIF)

**S31 Fig. Examples of 2D samples obtained from 3D simulated tumours.** 2D sampling of 3D simulated tumours with model parameters $s = 1$, $n_{mut} = 0.01$ **(a)** $q = 0$, **(b)** $q = 5$ and **(c)** $q = 10$. Samples are obtained by sweeping through the 3D tumour along three orthogonal axes (one direction of sampling depicted in 3D images on left of each sub-figure).
(TIF)

**S32 Fig. CMFPT analysis of 2D and 3D derived simulated sub-clonal patterns.** CMFPT analysis of sub-clonal patterns generated using 2D and 3D simulations for a range of sub-clonal parameters. CMFPT measurements are plotted in the $(\tilde{\tau}_{ry}, \tilde{\tau}_{ry}/\tilde{\tau}_{yr})$ phase space) with measurements of patterns derived from 3D tumours coloured according to their ratio of mutant to WT cell numbers, $\phi$. Measurements of sub-clonal patterns generated with the corresponding 2D system are coloured blue.
(TIF)

**S1 Table. Parameters used for 2D spatial simulations.**
(XLSX)

**S2 Table. Summary of all sub-sample best-fit parameters.** Best-fit model parameters, formatted as $(s, n_{mut}, q)$. Sub-sample number is shown in parentheses next to BaseScope sample number.
(XLSX)

## Author Contributions

**Conceptualization:** Magnus J. Haughey, Aleix Bassolas, Sandro Sousa, Vincenzo Nicosia, Weini Huang.

**Data curation:** Ann-Marie Baker.

**Formal analysis:** Magnus J. Haughey, Aleix Bassolas, Sandro Sousa.

**Investigation:** Magnus J. Haughey, Aleix Bassolas.

**Methodology:** Magnus J. Haughey, Aleix Bassolas, Vincenzo Nicosia, Weini Huang.

**Supervision:** Trevor A. Graham, Vincenzo Nicosia, Weini Huang.

**Validation:** Magnus J. Haughey.

**Visualization:** Magnus J. Haughey, Aleix Bassolas.

**Writing – original draft:** Magnus J. Haughey, Aleix Bassolas.

**Writing – review & editing:** Magnus J. Haughey, Aleix Bassolas, Sandro Sousa, Ann-Marie Baker, Trevor A. Graham, Vincenzo Nicosia, Weini Huang.

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
