## [Decision Letter · Decision Letter 0]

20 Jul 2022

Dear Mr Haughey,

Thank you very much for submitting your manuscript "First passage time analysis of spatial mutation patterns reveals sub-clonal evolutionary dynamics in colorectal cancer" for consideration at PLOS Computational Biology.

As with all papers reviewed by the journal, your manuscript was reviewed by members of the editorial board and by several independent reviewers. In light of the reviews (below this email), we would like to invite the resubmission of a significantly-revised version that takes into account the reviewers' comments.

In particular, the referees question the validity of the model assumptions. For example, the decision to model a 3D process in 2 dimensions likely has significant impact to the extent that it may affect the conclusions. Also the biological relevance of "cell pushing" is questioned in the rather confined environment in which these tumors grow. All of this made the referees ask if the same tumor structures could not also be reproduced based on a set of completely different model assumptions, or an extended set of plausible model assumptions, e.g., birth/death processes without cell pushing, non-automatous cell-cell interactions, a simple model of the immune system, possible three-dimensional simulations based on the same or similar assumptions, cell motility/cell mixing and so forth. Comparing predictions for different (potentially overlapping) sets of model assumptions should give more insight into this matter.

As for the CMFPT method, while the referees appreciate the novel application of this method, the claim that this is a 'better method' than other methods should be supported with thorough comparisons with existing methods for assessing pattern heterogeneity. Also the question is raised whether this method is not unnecessarily complex and if not faster and more simply defined methods (e.g.mean shortest distance) could be used instead. 

Further important issues include possible overselling of the clinical relevance at present stage of this work, and the role of resistance cost. Please address these and the other comments of the reviewers in a revised manuscript. 

We cannot make any decision about publication until we have seen the revised manuscript and your response to the reviewers' comments. Your revised manuscript is also likely to be sent to reviewers for further evaluation.

Sincerely,

Roeland M.H. Merks, Ph.D

Associate Editor

PLOS Computational Biology

Natalia Komarova

Deputy Editor

PLOS Computational Biology

Reviewer's Responses to Questions

**Comments to the Authors:**

Reviewer #1: The authors present an interesting manuscript that addresses the need for methods to analyze dynamic data for which there is only a single timepoint available. The authors utilize a method previously employed in other fields and consider its application to the study of cancer evolution.

By using computational simulations (for which the ground truth is well known) and clinical data, the authors provide evidence of the usefulness of their proposed method. While I think that this research is significant and worthy of publication I found some issues (some more serious than others). I don't expect to have these points addressed in the research per self but at least addressed in the discussion in ways that ensure that the reader is aware of the limitations of the work which, otherwise, remains novel and interesting.

The finding that pushing has a huge impact the topology seems relatively trivial if one considers that in this ABM a relatively similar effect could be obtained by adding cell motility.

The model is 2D but I think that while reality is 3D, much can be gained from 2D sections that reflect real evolutionary patterns. A different matter is the impact of the normal cells and the physical environment which is neglected by this work (admittedly and upfront). Contrary to what the authors claim this is no minor neglect, it is clear it will impact evolutionary dynamics (interactions between normal and tumor is not just about competition for resources and there are plenty of non cell-autonomous interactions) and while not easy, if they look in the literature they will find plenty of examples that demonstrate that this is possible and frequently done. Furthermore, while it is mentioned that inter patient variability could be quite high, it would seem to this reviewer that this variability might be a key factor that could explain different evolutionary trajectories between patients with similar genetic (original) makeup.

Please, in the revision embed the figures with the manuscript, it makes for much easier reading! Also, for some reason the quality of the main figures (but not of the supplementary ones) is very poor and the axis and legends are very hard to read.

Reviewer #2: There are some interesting ideas and a lot of work in this manuscript, combining theory and analysis of experimental data. However it does not convincingly shows any usefulness of the developed method. On the one hand, it states that this method is more detailed than another already existing method, but the manuscript does not show enough comparisons with existing methods, and one could think of other more straightforward methods which would similarly convey the spatial arrangement in simpler observables. On the other hand, there are several aspects of the method that are questionable, in particular the question of the number of cells considered, and that the information on the proportion of mutant area is not used; on top of the acknowledged issue of the 2 dimensional representation of the 3 dimensional process. For all these reasons, I recommend rejection of this manuscript despite some other qualities.

Below, more about the 2 main points; and then a list of smaller comments.

* Main point 1: not enough comparisons with existing methods

One comparison not directly made is with the previous method using Shannon's entropy. In the paragraph starting around line 386, "These analyses demonstrate how the CMFPT provides a more detailed description of the patterns of mutant sub-clonal mixing than the spatial Shannon’s entropy measure that was previously used to analyze the BaseScope patterns" : there is actually no direct comparison with previous results. "Whilst the patterns produced under a surface growth model and one with cell pushing were qualitatively similar when analysed using Shannon’s entropy, such dynamics were measurable when analysing the mixing patterns at multiple spatial scales using the CMFPT". If one looks at figure S6 where triangles represent values for experimental images, and figure S5 which shows the simulated points for each panel a different q value (large q values are strong cell pushing, q=0 is surface growth model), it is not clear that the experimentally observed regions of the taury, taury/tauyr space belongs more to a q value than another. Figure 5f shows the comparison between expected and inferred values of the various parameters for simulated data (much easier to compare than experimental data of course), and, while there is some signal, there is a relatively high error rate.

In the discussion, in the paragraphs between lines 426 and 462, the authors actually compare with another more sophisticated method, and while for some data there is some qualitative agreement between this manuscript analysis and this other study, there are disagreements with other data. So it is not very conclusive.

Besides, the main point is to represent with 2 observables (the rescaled tau r->y and the ratio of the rescaled tau r->y / tau y->r) the spatial distribution (homogeneous domains vs. more diffuse). It is done using a complicated metrics, the class mean first passage times (CMPFT). I have nothing a priori about calculating MFPT, but I don't really see how that conveys more information than metrics such as a mean shortest distance, which would be much easier to compute, and also easier to check its validity for known patterns, such as those generated in figure 2. It seems complex for no real gain.

* Main point 2: questionable aspects of the methods

The authors do acknowledge that the real process is 3-dimensional, while their simulations that they use as a model, to infer to which parameters the observables correspond to, is 2-dimensional. A 2-D slide of a 3-D process is indeed quite different from the outcome of a 2D process. To give an example, let's imagine 2 sheets, one with only WT, the other with only mutants, so a very segregated process. If both sheets are wrinkled, and there are on top of each other, an horizontal plane cutting through both sheets will get some of the bottom sheet, then some of the top sheet, then the bottom, then the top, and so on, giving the impression of a much more diffuse process. At least on this point the authors are aware of the potential limitation.

Around line 330, it is acknowledged that scale can be important, as when discussing the case with analysis of subparts of an image, it is stated that "(...) the perceived cell pushing strength may be most affected by the sub-sampling strategy employed here, as even highly mixed patterns will begin to appear segregated when viewed with a higher magnification". However nowhere there is any discussion of whether the right scale is used. The simulations are done for clusters stopping at Nmax=10^5 cells. Does it correspond to the right order of magnitude in the imaged tumors? How sensitive are the estimates to perform the simulations with different Nmax? Also, the experimental images are subsampled to 3000 pixels (images of 3000x3000?) (paragraph starting line 342). The simulations run for Nmax=10^5 in 2D, thus typically a final size of 300 cells x 300 cells, thus smaller than the subsampled images of the tumors. That the scales are not matching may be an issue for parameter estimation, and this is not at all explored.

In the section starting around line 587, what is explained is a method finding the simulations best matched with the experimental patterns, in regards to tau_rr, tau_ry, tau_yr, tau_yy. Why not using phi, the proportion of a given color, as an additional info? It's very easy to compute, and it is concerning than in figure 6, triangles representing experimental images have sometimes very different color (representing phi) than the simulated points to which they seem close to for the CMFPT values.

* smaller comments (more or less in the order they appear in the manuscript):

- the value of Nmax and other parameters should be in the legends of the main text, not to be dug up somewhere in the SI

- line 109: should be 0 < phi < 1, not the <= and >=, as the quantities are not defined for phi=0 and phi=1.

- I found it a bit confusing to use tmut as a name for the parameters representing the proportion of Nmax, which when reached, makes the simulation introduce a mutant. It is not directly a time, as the number of new cells in a given unit of time scales as the number of existing cells.

- Around line 166 "Due to the coupled effects of s and t mut on the final class ratio, tumors with early arising, weakly selected mutant sub-clones (Fig 4, small blue points), and those with later arising, strongly selected sub-clones (Fig 4, large red points) share similar values of phi (shown by colors in Fig 4a-d, inset). Interestingly, despite having a similar mutant frequency, phi, the CMFPT is sensitive to the topological differences between patterns generated in these two contrasting scenarios. Tumors with the earliest arising sub-clones occupy a unique area of the phase space for all cell pushing strengths." Figure 4 there are clearly point with warmer colors (larger s) and larger size (larger tmut) colocalized with points of colder colors and smaller size (smaller s, smaller tmut). So it is actually not really the case that the CMFPT is able to contrast these 2 scenarios.

- All the discussion around page 11 about the non-tumorous tissue. Removing it and selecting subparts by hand may introduce bias. Actually it is also removing additional information that could be used in a more sophisticated model.

- Could q be related to real mechanical properties of the samples?

- line 551 : stochastic birth : a stochastic birth rate may not be ideal for cells (this leads to an exponential distribution of duration between replications, with often very short times, whereas cells require a minimum time to be able to divide). That's an approximation often made, but that could be briefly discussed.

- equation (2) : (1+s)^k with k in {0,1} : why this definition? Why the need of putting k rather than just 1?

- around line 566 : note that this models gives more death when there is more replication. Why not, but this could be discussed as it is not self evident.

- figure 4 : the color of the small points cannot be seen.

- figure S1a : some scale bars are seen : which size do they correspond to?

Reviewer #3: I believe this is a potentially interesting contribution to our understanding of early cancer development and emergence of different subclones in space. However, I have some concerns related to the modelling choices, its impact, and possible overselling of the clinical impact.

1 Modelling choices

1A If I understand it correctly, there is no interaction of cells of different types assumed, beyond the cell pushing. It seems to be a strange assumption to me (the interactions more likely happen than do not happen in a situation where cells need nutrients to proliferate and survive and where space/resources are limited). The fact that we need cell pushing for division implies the space is limited, thus at least I would expect some sort of competition for space/resources instead of only pushing another cell when dividing. In that sense, I was surprised that this assumption was not listed among limitations of the modelling framework.

1B How does the fitness advantage relate to the common assumption of resistance cost, i.e. that therapy resistant cells proliferate slower in absence of treatment? And can there be a temporal fitness disadvantage, followed by a fitness advantage, for example? For example, can the fitness advantage happen only in a long-term once the resistant subclone outcompetes the sensitive one more effectively? And can you imagine the fitness advantage having a form of competitive advantage at all? Would it make a difference?

1C The cell pushing, especially adopted to a 2D lattice model, sounds very unnatural. In reality, the space is continuous and three-dimensional. Could you motivate why having the extension of Waclaw et al.'s algorithm makes your Agent-Based simulation more realistic and not less realistic? Isn't cell pushing you implemented in your 2D lattice model the main cause of the surprising results for patient C537? How could you validate whether this is the case?

1D I think exclusion or inclusion of other than cancer cells in the modelling should be motivated by type of cancer, its stage, and considered treatment. Therefore, I was rather surprised to not see immune cells in your model as you have an early stage of cancer with known strong involvement of immune cells. More discussion on this would be appreciated. (This however seems to be less critical than no interaction among different subclones.)

1E To perform inference of mutant replicative advantage (s), mutation timing (tmut) and

cell pushing strength (q) for each BaseScope sub-sampled region a grid-search was used. I do not understand why optimization was not used instead and how the specifics of the grid search were chosen (for example q=0,5, 10 or 20 - why not other values? Or did I miss it?). Could you elucidate on this?

2 Impact of the modelling choices

2A My belief is that at best, you provide a plausible hypothesis for emergence of subclones in space. I wonder what are other options, and how critical the choices you made (see part 1) are for your observation. What about the cell pushing being replaced by placing offspring to the nearest free space and including rather standard competition among different cell types, for example? Would that lead to the same outcomes? I have no intuition regarding this.

2B As the measurements are spatial, but include only small section of a tumor and 1 time point, many spatial models with enough parameters would fit this data well (enough). While I understand the choices you made, I question how could you validate your hypothesis better. Are there time-series data you could use, for example? Could discussion include thoughts on this, including alternative hypotheses?

2C Page 18, line 333: "Validation of our inference method (Fig. 5f) would suggest that the

majority of sub-regions across these samples were indeed consistent with weak cell

pushing, however it is also plausible that stronger cell pushing was acting within these

tumours, which led to the observed fragmented patterns of tumour cells, which in turn

necessitated the sub-region analysis in these samples." Aside of discussion on how realistic the pushing is when converted to the 2D lattice model, how would you validate this alternative hypothesis?

3 Possible overselling of clinical impact

The introduction starts with "Understanding the origins and effects of intra-tumour heterogeneity is a fundamental

challenge in cancer research and is critical for managing therapy resistance and

improving treatment outcomes."

3A As you consider only subclonal resistance (and not e.g. epigenetics, phenotypic switching) and as you assume that all resistance is pre-existent, your model would be valuable for cancers and treatment where this assumption is satisfied. Can you elaborate on how realistic these assumptions are, in general and for colorectal cancer?

3B Are you sure that understanding the specific mechanisms of resistance evolution help us to handle the worst cases, like metastatic cancers? From recent advances in evolutionary therapies in metastatic cancers I got an impression that it does not matter much how resistance occurs as long as we can address it strategically. Also, knowing how precisely the resistance emerged is I think impossible and likely not relevant for these advanced cancers. What am I missing?

In view of 3A and 3B, I would tune the statements on clinical implications (few in the paper) down.

4 Others

4A I believe one should understand a submission without the need to study other papers. Here I needed to read reference 20 to understand the scope. I would suggest to summarize earlier results, also including modelling, so that no reader needs to read other papers to understand your contributions.

4B Page 23, last line: "In line with contemporary models of tumour evolution..." Are you sure that your results are really in line with all contemporary models of tumor evolution? Again, tuning this statement down would be better.

4C (minor) Page 20, line 403: Missing table number.

**Have the authors made all data and (if applicable) computational code underlying the findings in their manuscript fully available?**

Reviewer #1: Yes

Reviewer #2: Yes

Reviewer #3: Yes

PLOS authors have the option to publish the peer review history of their article (what does this mean?). If published, this will include your full peer review and any attached files.

Reviewer #1: No

Reviewer #2: No

Reviewer #3: No
---

## [Decision Letter · Decision Letter 1]

15 Nov 2022

Dear Mr Haughey,

Thank you very much for submitting your manuscript "First passage time analysis of spatial mutation patterns reveals sub-clonal evolutionary dynamics in colorectal cancer" for consideration at PLOS Computational Biology.

As with all papers reviewed by the journal, your manuscript was reviewed by members of the editorial board and by several independent reviewers. In light of the reviews (below this email), we would like to invite the resubmission of a significantly-revised version that takes into account the reviewers' comments.

Reviewer 2 performed a check on the data in Figure S3 (see review and the attachment) and concluded that there are mistakes. I would like to give you an opportunity to write a detailed response to their concern and comment on its implications for the rest of the results. I will likely send your response back to the reviewer for advice, based on which I will make a final decision.

We cannot make any decision about publication until we have seen the revised manuscript and your response to the reviewers' comments. Your revised manuscript is also likely to be sent to reviewers for further evaluation.

Sincerely,

Roeland M.H. Merks, Ph.D

Academic Editor

PLOS Computational Biology

Natalia Komarova

Section Editor

PLOS Computational Biology

Reviewer's Responses to Questions

**Comments to the Authors:**

Reviewer #1: I believe that the authors have addressed, at least in terms of clarifications in the manuscript, the concerns I raised in my review

Reviewer #2: The authors did an extensive revision work. However, I am really concerned that the results that I can easily check are not ok. Given that an easy calculation is false, I have trouble trusting the rest, which is harder to thoroughly check.

See in the attached document the calculation for the figure S3. Figure S3 is analogous to figure 2 of the main text, but with the mean shortest distance instead of the CMFPT. The shortest distance is easier to compute than the CMFPT. In the column and small cluster cases (most convincingly for the column case), it is possible to calculate the mean shortest distance, and it does not correspond to what is shown on the right panel of figure S3.

**Have the authors made all data and (if applicable) computational code underlying the findings in their manuscript fully available?**

Reviewer #1: Yes

Reviewer #2: None

PLOS authors have the option to publish the peer review history of their article (what does this mean?). If published, this will include your full peer review and any attached files.

Reviewer #1: No

Reviewer #2: No
---

## [Decision Letter · Decision Letter 2]

4 Jan 2023

Dear Mr Haughey,

Thank you very much for submitting your manuscript "First passage time analysis of spatial mutation patterns reveals sub-clonal evolutionary dynamics in colorectal cancer" for consideration at PLOS Computational Biology. As with all papers reviewed by the journal, your manuscript was reviewed by members of the editorial board and by several independent reviewers. The reviewers appreciated the attention to an important topic. Based on the reviews, we are likely to accept this manuscript for publication, providing that you modify the manuscript according to the review recommendations.

Sincerely,

Roeland M.H. Merks, Ph.D

Academic Editor

PLOS Computational Biology

Natalia Komarova

Section Editor

PLOS Computational Biology

Reviewer's Responses to Questions

**Comments to the Authors:**

Reviewer #2: First, I wish to thank the authors for the explanation. I had missed that for the shortest distance, it is also renormalized, and I find it very reassuring to see that then the simulation results are as expected from analytics in the simple cases for which it is possible to calculate what is expected.

Going back to the rest of the 1st re-submission, I thank the authors for their extensive revision work. There are some interesting additions to the manuscript. I would say that I am still not fully convinced of the usefulness of the method proposed compared to alternatives, but given that the claims are toned down compared to the first draft, and that some points are now clearer, I think that this exploration of this new method is worth publishing. I have a few more comments, but small revisions are enough.

Here, the detailed comments in 3 parts. First, the elements which makes me unconvinced, and thus justifying to tone down a bit more the text. Second, some specific elements that still feel like overselling. Third, other small comments, more or less in the order of the text.

1/ First, about why I am not totally convinced by the answers to my concerns:

In the response to the reviews, in regard to the CMFPT vs. shortest distance, the authors state that "We agree with the reviewer and show that the results using mean shortest distance are similar to those derived using the CMFPT (Supplemental figures 3 and 4 in the revised manuscript). We argue, however, that the CMFPT conveys slightly more information about the pattern in question, due to the process of finding the distribution of distances from cell type A to B using a stochastic process, compared to the deterministic process of finding the shortest distance (i.e. the approach of the mean shortest distance metric). This means that the CMFPT enables us to obtain a distribution of transition times for every starting node in the network, unlike the mean shortest distance, and therefore conveys more information about the neighborhood of the starting cell." Indeed to calculate the CMFPT, there are many trajectories calculated, and thus a distribution. But then in the comparison with simulations of the process, only the mean is computed. Thus no more information is conveyed, though the information conveyed is slightly different. One would need another measure to compare with the simulations, for instance the variance of the first passage time, to really use the distribution. And indeed in the simple examples, there is very little difference in the patterns between CMFPT and shortest distance. Given that using the CMFPT instead of the shortest distance does not seem to affect much how well patterns can be distinguished (see figure 2 vs figure S3; and figure 4 vs. figure S4), using a technique that is more computationally intensive and potentially noisy, is not clearly justified.

The answer about scales is not really satisfying, because actually (sometimes small) subparts of the tumors are compared, not the whole image of the 8000micronsx8000microns field. Thus comparing bits of different sizes to always simulations with the same N total does not seem right. It is actually less important than the 3D vs. 2D issue (which also make for different Ntot). I also do not understand the (arbitrary?) choice of subparts of the tumors to be compared. Indeed, the results are showing different inferred parameters within the same slice.

The simulations with similar rescaled CMFPT give sometimes very different patterns. See for instance figure 6: [1] : phi is completely off. [4] and patterns on the left: the yellow area much more fragmented in real samples than in the simulation: could be strong indication of multiple mutations for instance, or transport not consistent with model used here.

Finally, the manuscript makes clear a series of limitations (2D vs 3D, not using for now the additional information of phi, limit in the number of simulated parameter values (e.g. q), dynamics of the non-cancerous cells omitted) which will make hard to apply such a method, but these explanations of the limitations are actually a good point of the present manuscript.

2/ Here a list of places where the claims should be toned down:

- in the abstract "We uncover a wide range of sub-clonal dynamics" > "We infer a wide range of sub-clonal dynamics" (there is no guarantee that the results are actually what happened in the tumor)

- around line 148: "Mean shortest distance is a deterministic measure of the distance between cells, whereas CMFPT constructs paths between pairs of cells using a stochastic process, resulting in a distribution of first passage times for each starting cell. As such, CMFPT provides more information about the texture of the patterns than mean shortest distance by incorporating information about the neighbourhood of each cell, and gives a better description in more complex patterns and experimental data." Given only the mean is used, and that the supplementary figure with the shortest distance is very similar to the one wit the CMFPT, there is no clear reason that "more" information is given. The last sentence could be changed to something like "As such, CMFPT may provide a different information about the texture of the patterns than the mean shortest distance, by incorporating information about the neighborhood of each cell. "

- around line 226: "it is possible to recover details about the underlying cell by quantifying the resulting spatial patterns alone." : a bit too optimistic, as there are several sets of parameters giving similar results: "it is possible to narrow down the potential underlying cell dynamics by quantifying the resulting spatial patterns alone."

- After figure 6 there should be at least a sentence commenting that some patterns of the simulations are actually not consistent with the actual sample, despite having similar CMFPT.

3/ Other comments

- how many trajectories to compute the CMFPT?

- around line 75 : why mention tau yy and tau rr? Never used afterwards, and their definition will be more sensitive to the chosen subsampled spatial grid.

- legend of figure 4: no description of the smaller figures

- figure 4 : I suppose the smaller figures show a point for each simulation, with 100 simulations for each symbol of the larger figure, (with color for the final phi). There seem to be quite some spread in some cases. It would be interesting to show the typical range of outcomes for a given q,s,nmut, for instance with errorbars on the symbols. That may be too much for this figure that is already busy, but one may want to keep this in mind when comparing simulations and experimental images.

**Have the authors made all data and (if applicable) computational code underlying the findings in their manuscript fully available?**

Reviewer #2: **No: **"Data analyzed in this study are available from the corresponding authors upon reasonable request." Not acceptable for plos. Anonymized data, and only a few images, at sub-resolution of 3000x3000, that should not be a big burden to make more accessible.

PLOS authors have the option to publish the peer review history of their article (what does this mean?). If published, this will include your full peer review and any attached files.

Reviewer #2: No

Figure Files:

Data Requirements:

Reproducibility:

References:

---

## [Decision Letter · Decision Letter 3]

14 Feb 2023

Dear Mr Haughey,

We are pleased to inform you that your manuscript 'First passage time analysis of spatial mutation patterns reveals sub-clonal evolutionary dynamics in colorectal cancer' has been provisionally accepted for publication in PLOS Computational Biology.

Best regards,

Roeland M.H. Merks, Ph.D

Academic Editor

PLOS Computational Biology

Natalia Komarova

Section Editor

PLOS Computational Biology

Reviewer's Responses to Questions

**Comments to the Authors:**

Reviewer #2: Thanks for answering all the points.

**Have the authors made all data and (if applicable) computational code underlying the findings in their manuscript fully available?**

Reviewer #2: Yes

PLOS authors have the option to publish the peer review history of their article (what does this mean?). If published, this will include your full peer review and any attached files.

Reviewer #2: No

---

## [Editor Report · Acceptance letter]

6 Mar 2023

PCOMPBIOL-D-22-00569R3 

First passage time analysis of spatial mutation patterns reveals sub-clonal evolutionary dynamics in colorectal cancer

Dear Dr Haughey,

I am pleased to inform you that your manuscript has been formally accepted for publication in PLOS Computational Biology. Your manuscript is now with our production department and you will be notified of the publication date in due course.

With kind regards,

Anita Estes
